# Evaluating the PurpleAir monitor as an aerosol light scattering instrument

James R. Ouimette[1], William C. Malm[2], Bret A. Schichtel[3], Patrick J. Sheridan[4], Elisabeth Andrews[4,5], John A. Ogren[6], W. Patrick Arnott[7]

[1]Sonoma Ecology Center, Eldridge, CA 95431, USA
[2]Cooperative Institute for Research in the Atmosphere, Colorado State University, Fort Collins, CO 80523, USA
[3]National Park Service Air Resources Division, Lakewood, CO 80225, USA
[4]NOAA Global Monitoring Laboratory, Boulder, CO 80305, USA
[5]Cooperative Institute for Research in Environmental Sciences, University of Colorado, Boulder, Colorado 80309, USA
[6]NOAA Global Monitoring Laboratory, Boulder, CO 80305, USA (Retired)
[7]Department of Physics, University of Nevada, Reno, NV 89557, USA
Correspondence to  James R. Ouimette (JamesOuimette@sonomaecologycenter.org)

## Abstract

The Plantower PMS5003 sensors (PA-PMS) used in the PurpleAir (PA) monitor PA-II-SD configuration are equivalent to cell-reciprocal nephelometers using a 657 nm perpendicularly polarized light source that integrates light scattering from 18 to 166 degrees. Yearlong field data at the National Oceanic and Atmospheric Administration's (NOAA) Mauna Loa Observatory (MLO) and Boulder Table Mountain (BOS) sites show that the 1 h average of the PA-PMS first size channel, labeled ">0.3 μm" ("CH1") is highly correlated with submicrometer aerosol scattering coefficients at the 550 nm and 700 nm wavelengths measured by the TSI 3563 integrating nephelometer, from 0.4 Mm$^{-1}$ to 500 Mm$^{-1}$. This corresponds to an hourly average submicrometer aerosol mass concentration of approximately 0.2 to 200 μg m$^{-3}$. A physical-optical model of the PA-PMS is developed to estimate light intensity on the photodiode, accounting for angular truncation of the volume scattering function as a function of particle size. The model predicts that the PA-PMS response to particles >0.3 μm decreases relative to an ideal nephelometer by about 75% for particle diameters ≥1.0 μm. This is a result of using a laser that is polarized, the angular truncation of the scattered light, and particle losses (e.g., due to aspiration) before reaching the laser. It is shown that CH1 is linearly proportional to the model-predicted intensity of the light scattered by particles in the PA-PMS laser to its photodiode over 4 orders of magnitude.  This is consistent with CH1 being a measure of the scattering coefficient and not the particle number concentration or particulate matter concentration. The model predictions are consistent with data from published laboratory studies which evaluated the PMS against a variety of aerosols.  Predictions are then compared with yearlong fine aerosol size distribution and scattering coefficient field data at the BOS site. Field data at BOS confirm the model prediction that the ratio of CH1 to the scattering coefficient would be highest for aerosols with median scattering diameters <0.3 μm. The PA-PMS detects aerosols smaller than 0.3 μm diameter in proportion to their contribution to the scattering coefficient. The results of this study indicate that the PA-PMS is not an optical particle counter and that its six size fractions are not a meaningful representation of particle size distribution. The relationship between the PA-PMS 1 h average CH1 and b$_{sp1}$, the scattering coefficient in Mm$^{-1}$ due to particles below 1 μm aerodynamic diameter, at wavelength 550 nanometers, is found to be b$_{sp1}$ = 0.015 ± 2.07 × 10$^{-5}$ ×

CH1, for relative humidity below 40%.  The coefficient of determination $r^2$ is 0.97. This suggests
that the low-cost and widely used PA monitors can be used to measure and predict the submicron
aerosol light scattering coefficient in the mid-visible nearly as well as integrating nephelometers.
The effectiveness of the PA-PMS to serve as a $PM_{2.5}$ mass concentration monitor is due both to
the sensor behaving like an imperfect integrating nephelometer and to the mass scattering
efficiency of ambient $PM_{2.5}$ aerosols being roughly constant.

Keywords: PurpleAir, Plantower PMS5003, nephelometer, low-cost sensor, physical-optical
model, $PM_{2.5}$, scattering coefficient, visibility, atmospheric aerosol

## 1. Introduction

Currently there are tens of thousands of low-cost aerosol monitors used by atmospheric research
groups, air quality monitoring and regulatory organizations, and individual citizen scientists
around the world. The recent explosion in the number of these sensors (see, for example, Aerosol
Air Qual. Res. 20(2), 2020, Special Issue on Low-cost Sensors for Air Quality Monitoring and
papers therein) is a result of the increased research, regulatory, and citizen interest over the past
few years. For example, there are over 9,000 active PurpleAir (PA) aerosol monitors (PurpleAir
LLC, Draper, UT), with sampling locations on almost every continent. The large geographic
coverage of this array of low-cost sensors presents enormous potential for obtaining valuable
information on atmospheric aerosol properties and transport processes.

The majority of these low-cost aerosol sensors are used to monitor the mass concentration of
particles with aerodynamic diameters <2.5 μm ($PM_{2.5}$) (Kelly et al., 2017; Gupta et al., 2018;
Zheng et al., 2018; Sayahi et al., 2019; Barkjohn et al., 2020; Holder et al., 2020; Jayaratne et al.,
2020; Malings et al., 2020; Mehadi et al., 2020). However, these sensors do not actually measure
aerosol mass concentrations but light scattered by the aerosols and thus are dependent on the
aerosol particle size distribution, morphology, and composition.  Recently, Hagan and Kroll
(2020) developed a framework and computer model to estimate the effects of relative humidity
(RH) and aerosol refractive index on $PM_{2.5}$ estimated by a number of low-cost sensors. Their
model assumed that the low-cost sensor lasers were not polarized and could be modeled with
Mie theory. The PMS5003 (PMS) was included in their classification scheme as an example of a
sensor that behaved more like a nephelometer than an optical particle counter.

Three recent laboratory studies showed that the PMS response decreases with particle size.  He et
al. (2020) measured the PMS response to monodisperse ammonium sulfate aerosol particles
having diameters of 0.1, 0.3, 0.5, and 0.7 μm. The PMS was able to detect 0.1 μm particles. They
derived a transfer function that showed that the PMS >0.3 μm channel (CH1) response was
maximum at particle diameter 0.26 μm but decreased significantly below this size. They
concluded that the PMS behaved more like a nephelometer than an optical particle counter.
Kuula et al. (2020) generated monodisperse dioctyl sebacate oil droplets from 0.5 to 20 μm and
measured the PMS CH1 response versus particle diameter using an aerosol particle sizer (APS).
Their data showed that the PMS relative response decreased for particles >0.5 μm diameter.
Tryner et al. (2020) evaluated three low-cost particulate matter sensors, including the PMS, by
exposing them to five different types of aerosols in the laboratory.  They found that the ratios of
PMS-reported to filter-derived $PM_{2.5}$ mass concentrations were inversely proportional to mass
median diameter (MMD). Wood smoke had the smallest MMD, 0.42 μm; its PMS $PM_{2.5}$

averaged 2.5 times the filter-derived $PM_{2.5}$. Conversely, oil mist had the largest MMD, 2.9 μm;
its PMS $PM_{2.5}$ averaged only 0.23 times the filter-derived $PM_{2.5}$.
Climate modeling requires a robust set of models and atmospheric measurements for predicting
anthropogenic aerosol radiative forcing. Currently, there are uncertainties in the modeling
results, due in part to the sparseness of ground-based data used to evaluate and refine the models
(e.g., Gliss et al., 2021). Satellite observations provide global coverage that can be used for
model evaluation, but satellite data require further assessment, particularly when trying to
provide information about surface aerosol properties. The Surface Particulate Matter Network
(SPARTAN) (https://www.spartan-network.org/; Snider et al., 2015) was specifically designed
to assess and improve algorithms to relate satellite retrievals to surface aerosols. SPARTAN
operates collocated filter-based $PM_{2.5}$, aerosol scattering coefficient via nephelometer, and
aerosol optical depth (AOD) measurements at approximately 20 sites around the world. Model
and satellite uncertainties can be reduced using a distributed set of low-cost sensors that can
provide aerosol light scattering estimates at a higher spatial and temporal resolution than is
possible using nephelometers alone. Low-cost sensors are increasingly being used along with
satellite data to estimate global aerosol impacts (Gupta et al., 2018).
There is ongoing scientific debate about the accuracy and precision of these low-cost sensors and
their limitations (Morawska et al., 2018; Jayaratne et al., 2020). Many of the recent papers
discuss performance evaluations or "calibrations" of these low-cost sensors by comparing their
measurements with traditional, research-grade aerosol measurements (Papapostolou et al., 2017;
Barkjohn et al., 2020). The concerns over data quality, stemming largely from inexpensive
components, lack of transparency of signal processing, and inadequate quality control and testing
at the factory, must be weighed against the advantages of low cost and wide spatial coverage.
The actual measurement in the PA monitor with its two PMS5003 sensors (PA-PMS), and in
many other low-cost aerosol monitors, is of light scattered by particles integrated over a wide
range of angles (Kelly et al., 2017), which has traditionally been done in atmospheric research
and aerosol monitoring programs using integrating nephelometers. Aerosol light scattering and
extinction measurements are useful in many applications, including determination of the
radiative forcing effects of aerosols on climate change, atmospheric visibility, wildfire/smoke
impacts, and validation of model outputs and satellite retrievals (e.g., Malm et al., 1994;
Sherman et al., 2015; Snider et al., 2015; Gliss et al., 2021). Even though most low-cost aerosol
sensors use light scattering as the basis of their operation, almost none have been evaluated as a
low-cost nephelometer to estimate atmospheric light scattering. Markowicz and Chilinski (2020)
conducted a 3 year evaluation of two low-cost sensors versus the Aurora 4000 polar integrating
nephelometer at a site in southeastern Poland. They found that the mass concentration of
particles with aerodynamic diameters <10 μm ($PM_{10}$) from the DfRobot SEN0177 and the
Alphasense OPC-N2 were highly correlated ($r^2 > 0.89$) with the aerosol scattering coefficient
measured by the nephelometer. They were able to estimate the 1 h average aerosol scattering
coefficient from the low-cost sensors with a root mean square error (RMSE) of 20 $Mm^{-1}$,
corresponding to 27% of the mean aerosol scattering coefficient.
Unfortunately, due to cost, availability, and the expertise required to run them, integrating
nephelometers are not operated in great numbers around the world. A recent analysis by Laj et
al. (2020) showed 56 long-term monitoring stations reporting their nephelometer data to the
World Meteorological Organization (WMO) Global Atmosphere Watch (GAW) World Data
Centre for Aerosols. This count includes nephelometers operated in several monitoring
networks, including the National Oceanic and Atmospheric Administration's (NOAA) Federated
Aerosol Network (NFAN, Andrews et al., 2019), the Aerosols, Clouds and Trace Gases Research
Infrastructure (ACTRIS) network (e.g., Pandolfi et al., 2018), and the Interagency Monitoring of
Protected Visual Environments (IMPROVE) network (Malm et al., 1994). While there are more
nephelometers in use around the world for short-term field and laboratory studies, the number
almost certainly does not exceed a few hundred. This is small compared with the number of low-
cost aerosol monitors in use globally.
This paper presents an evaluation of the performance characteristics of the low-cost PA-PMS
monitor to measure the integrated aerosol light scattering coefficient. It is shown that the PA-
PMS sensor configuration is similar to a cell-reciprocal nephelometer. A physical-optical model
based on Mie theory and the PMS geometry is created that predicts scattered light intensity on
the PMS photodiode and aerosol forward and backward light scattering truncation. PA-PMS
measurements are compared to yearlong measured aerosol light scattering coefficients at
NOAA's Mauna Loa Observatory (MLO) in Hawaii and to measured and modeled aerosol light
scattering coefficients and aerosol size distribution at the Boulder Table Mountain (BOS) site in
Colorado. Finally, an empirical relationship is developed to estimate the submicron light
scattering coefficient and its uncertainty from the PA-PMS data.
With a better understanding of what the PA measures, how it works, and its uncertainties, the
large network of PA-PMSs could be used to estimate the submicrometer aerosol scattering
coefficient at visible wavelengths throughout the world. These data could then be used to
improve chemical transport and general circulation models, advance climate change predictions,
and provide for better air quality forecasts.

## 2. Instrument description

In this section we first describe the physical and optical characteristics of the PA-PMS to place it
in the context of nephelometry. We then provide a brief overview of integrating nephelometers,
which are instruments designed specifically to measure light scattering.

### 2.1 PA-PMS nomenclature

The PMS sensor outputs 14 fields that are processed and reported by the PA. Each of these
fields will be referred to as a channel. For instance, the PA-reported number concentration of
particles >0.3 μm is referred to as CH1 in the remainder of this paper, number concentrations
>0.5 μm as channel two (CH2), and so forth. Furthermore, the two PMS sensors embedded in
the PA will be referred to as either sensor A or sensor B. Therefore, the number concentration of
particles >0.3 μm derived from sensor A will be referred to as CH1A and those from sensor B as
CH1B. The average of CH1A and CH1B will be referred to as CH1avg. The PMS reports the
CH1 units as "#/dl", which is the number of particles having diameters >0.3 μm per deciliter. In
this paper the PMS units for CH1 are not used.

### 2.2 Description of the PA and its PMS 5003 sensors

The PA monitor integrates two PMS sensors, a Bosch BME280 pressure, temperature, and RH
sensor and an ESP 8266 chip (https://www2.purpleair.com/pages/technology). The PA-reported
temperature and RH are based on the sensor attached to the circuit board, and do not necessarily
represent ambient conditions. The available specifications of the PMS are incomplete, and the
processing algorithms are unknown (He et al., 2020). The following is based on available
information and, where needed, professional judgment.  Each PMS includes a small laser, a
photodiode, a small fan to draw air across the laser beam, a microprocessor control unit (MCU),
and probably an operational amplifier. The MCU processes the signal from the photodiode and
outputs the following data fields approximately once per second:  >0.3 μm, >0.5 μm, >1.0 μm,
>2.5 μm, >5 μm, >10 μm, $PM_1$, $PM_{2.5}$, and $PM_{10}$. The PMS denotes the first six data fields as
particle number concentrations above the designated cutpoint and the last three data fields as
mass concentrations of particles below the designated cutpoints; the PM data fields are reported
for two different conditions, "standard particles" and "under atmospheric environment". The PA
ESP8266 chip calculates 2 min averages of the PMS and BME280 signals. It transmits them
wirelessly and writes them as a CSV file on a micro SD card.

### 2.2.1 Airflow and particle losses

The recommended orientation of the PA results in aerosol being drawn upward by a small fan
through four 3 mm diameter entrance holes in each PMS. The aerosol then enters a 9.4 $cm^3$
chamber (Fig. S1a) and flows upward, parallel to and exposed to the circuit board as shown in
Fig. S1b. Particles then make a 180 degree turn through three exit holes at the top of the chamber
to emerge on the other side of the circuit board and flow downhill through a 1.1 $cm^3$ channel that
is illuminated by the laser. The total PMS volume is estimated to be 9.4 + 1.1 = 10.5 $cm^3$. The
PMS volumetric flow rate is estimated to be 1.5 $cm^3 s^{-1}$ (~0.090 lpm) based on measurements
described in Supplemental Materials Sect. S1. The estimated inlet velocity through the entrance
holes is estimated to be 5.3 $cm s^{-1}$.
The PMS inlet orientation 90 degrees to the wind, upward flow, and the low inlet velocity
through the sampling holes can result in significant aspiration losses of larger particles (Hangal
and Willeke, 1990). Aspiration losses are greater at higher wind speeds because it is more
difficult for the larger particles to follow the streamlines into the low velocity PMS inlet. This
can result in a lower concentration of larger particles entering the PMS than are in the ambient
air.  Particle aspiration losses are proportional to the particle Stokes number and the ratio of the
wind velocity to the inlet face velocity (Hangal and Willeke, 1990). More details are provided in
Supplemental Materials Sect. S1.
At typical wind velocities of 1–3 $m s^{-1}$, the ratio of PMS inlet face velocity to wind speed is only
0.02 to 0.05, much lower than typical sampling ratios of 0.5 to 6.0 (Brockman, 2011). Pawar and
Sinha (2020) addressed this problem for the Laser Egg low-cost sensor by putting it in a box and
adding a 40 lpm fan to increase the inlet-to-wind velocity ratio and to direct the airflow upward
to the Laser Egg inlet. During calm winds, large particle aspiration losses may occur by particle
gravitational settling, acting against the PMS upward flow (Grinshpun et al., 1993). The actual
wind conditions in the ambient air and in the PA near the PMS sample inlet are turbulent. Hangal
and Willeke (1990) found in their wind tunnel experiments that turbulence intensity had a
negligible effect on aspiration efficiency. Calculations using Eq. S1 (see Fig. S2) predict that at a
wind speed of 1 $m s^{-1}$, the PMS aspiration losses for particles >2 μm may be significant.
However, it must be cautioned that the literature does not include data for the very low 5.3 $cm s^{-1}$
PMS face velocity, and actual measurements of the PMS aspiration efficiencies were not made.
They may be significantly different from these calculated efficiencies.
Inside the PMS 9.4 cm$^3$ chamber, the air has an average velocity of 0.57 cm s$^{-1}$ and Reynolds
number of 6.1, resulting in an average residence time of 6.3 s.  The average air velocity in the
chamber is equal to the sedimentation velocity of a spherical 10 μm diameter particle with a
density of 2 g cm$^{-3}$ in air at STP (standard temperature and pressure; values used in this analysis
are 273.15 K and 1013.25 hPa, respectively). This suggests that some 2 g cm$^{-3}$ density particles
with diameters >10 μm that enter the PMS would settle out in the chamber and not make it to the
three exit holes at the top of the chamber. Ultrafine particles can also be lost to the walls of the
chamber and the printed circuit board due to convective diffusion. Calculations using the
equation for diffusional losses (Friedlander, 1977) show that less than 1% of the 0.01 μm
diameter aerosols would be lost in the chamber due to convective diffusion, with even smaller
diffusional losses for larger particles.
Loss of particles due to inertial impaction on the wall opposite the three holes (Fig. S1b) was
estimated by the local air flow Reynolds number near the three holes and the aerosol Stokes
number.  The local Reynolds number is calculated to be 23, and the Stokes number for 10 μm
particles is $8.2 \times 10^{-4}$.  At these low numbers, the calculated loss to impaction is less than 1% for
all particles less than 10 μm diameter (Hering, 1995).
The average flow velocity through the laser beam is approximately 3.0 cm s$^{-1}$.  By the time the
air flows through the laser beam, it has lost most of the particles over 10 μm diameter. Further
particle losses due to gravitational settling over the photodiode would be very small, since the
gravitational force is parallel to the photodiode.
In summary, it is likely that the laser in the PMS is sampling a lower concentration of particles >
2 μm diameter than in the ambient air. Based on the literature and calculations, the dominant
coarse aerosol loss mechanism may be aspiration, not internal losses. However, further
measurements are needed to assess the various aerosol loss mechanisms.
**2.2.2 Laser**
The wavelength and power of three PMS diode lasers were measured using an Ocean Optics Red
Tide USB650 spectrometer and Melles Griot Universal Optical Power Meter, respectively. The
wavelength averaged 657 +/- 1 nm, and the power averaged 2.36 +/- 0.04 mW.  The laser is
polarized parallel to the plane of the photodiode detector. This results in the aerosol-scattered
light being polarized perpendicular to the plane of incidence.  Figure S3 shows that
perpendicular polarization results in significantly greater scattering intensity from 0.3 μm
particles compared to natural or parallel polarization. It is probable that many low-cost PM
sensors have lasers that are polarized. Polarization will affect how the sensors respond to various
size particles and needs to be considered when modeling sensor behavior.
The PMS laser beam profile is not a simple plane wave, but complex in shape. The laser has a 3
mm diameter lens that focuses the laser over the photodiode. The beam profile evolves
significantly as it goes through the focal region (Naqwi and Durst, 1990).  The laser beam
diameter in the laser sensing region over the photodiode was not measured.  It was estimated by
eye to be 0.5 to 1.0 mm, with significant uncertainty.  The PMS MCU turns the laser on and off
every 800 msec or 2.5 s, depending on aerosol concentration. The laser pulses are 600–900
msec, with the laser power on continuously during this time. We hypothesize that the PMS MCU
gathers data during laser on, processes it during laser off, and uses the difference of the
photodiode output during these stages to obtain and subtract any electronic or stray light (other
than the laser) background signal to the photodiode.

### 2.2.3 Photodiode detector

The actual photodiode model in the PMS is unknown. The photodiode appearance is similar to
the BPW34 silicon PIN photodiode. In this paper the specifications of the BPW34 are used to
estimate the likely properties of the detector in the PMS. It has a very large dynamic range when
operated with reverse bias. The dependence of the photodiode current on the light intensity is
very linear over 6 or more orders of magnitude, e.g., in a range from a few nanowatts to tens of
milliwatts. Silicon PIN photodiodes have low dark current, a 20 nanosecond rise time, and good
wavelength sensitivity between roughly 400 and 1000 nm. (https://www.rp-
photonics.com/photodiodes.html). At a wavelength of 657 nm, the BPW34 produces
approximately 0.4 microampere current per microwatt of incident radiant power
(https://www.fiberoptics4sale.com/blogs/archive-posts/95046662-pin-photodetector-
characteristics-for-optical-fiber-communication). The PMS does not have any optical elements
to capture and focus the aerosol-scattered light on its photodiode.
The photodiode does not have a cosine corrector in front and is probably not a true cosine
detector. However, the relative spectral sensitivity is advertised to be a cosine response by the
manufacturers
(https://www.osram.com/ecat/DIL%20BPW%2034%20B/com/en/class_pim_web_catalog_1034
89/prd_pim_device_2219537/ and https://www.vishay.com/docs/81521/bpw34.pdf).

### 2.2.4 Laser and photodiode geometry

The PMS geometry is very similar to a cell-reciprocal nephelometer. Figure 1 shows the PMS
laser and photodiode geometry. The measurements were made with a Brown & Sharpe
micrometer. The distance from the laser exit hole to the photodiode is 2.5 mm; the perpendicular
distance from the center of the laser beam to the photodiode is 1.8 mm; the diameter of the
exposed photodiode area is 3.0 mm; the thickness of the base mask over the photodiode is 0.46
mm; and the distance from the edge of the photodiode to the end of the laser sensing volume is
4.5 mm. $\theta_1$ is the lower angular scattering limit, and $\theta_2$ is the upper angular scattering limit for a
particle in the laser.
Due to the PMS geometry, the upper and lower angular scattering limits for $\theta$ depend on the
location, x, of a particle in the laser. This can be seen in Fig. S4. For example, at x = 0 mm, at the
laser exit, the upper and lower scattering limits for $\theta$ are 18–38 degrees. At x = 4.0 mm, over the
center of the photodiode, the angular integration limits are 50–130 degrees. The PMS photodiode
is not capable of detecting light scattered from particles at less than 18 degrees.
Figures S5–S9 provide more detail about the PMS dimensions and geometry.

### 2.2.5 PMS5003 sensing volume

The sensing volume is the volume in which the aerosol is irradiated by the laser. The sensing volume extends the length of the laser where the aerosol flows through it, approximately 10 mm. The sensing volume is shown in Fig. S9. The average residence time of a particle in the laser beam is approximately 30 ms. Some of the scattered light is detected by the photodiode and creates a voltage pulse approximately 30 ms wide. It appears that the photodiode is detecting either a cloud of particles from the sensing volume or individual pulses, depending on the concentration. At low concentrations, the aerosol concentration within the sensing volume is unlikely to be uniform, resulting in large relative changes in output per second.

### 2.2.6 Signal processing and electronics

It is not reported how the PMS MCU differentiates and processes the photodiode signals. The PMS MCU sends the PA a signal approximately every second in the form of a digital sequence of unsigned 16 bit binary data words, and CH1 is thought to be proportional to the photodiode current. The photodiode current was not measured in this study. The PA creates 80 s (Firmware Version 3) or 120 s (Firmware Version 4 and higher) averages and writes them to its micro SD card. We measured an average percentage difference of 0.3% between the 2 min averages reported by the PA and the 2 min averages calculated from the 1 s values from the PMS. The results are shown in Fig. S10. It is apparent that the processing done by the PA to calculate its reported 2 min averages does not bias the results.

### 2.2.7 PMS CH1 variability in sampling filtered air

We found significant variability in PMS response to filtered air. We exposed 21 PAs containing 42 PMS sensors to filtered air for 2 to 94 hours. The results are summarized in Table S1. Hourly average CH1 ranged from 0.10 to 377. Eleven PAs had both PMS CH1A and CH1B averages below 2, while seven PAs had at least one CH1 average over 26. We recommend that before deployment the PAs sample filtered air for at least four hours to identify and eliminate PAs with CH1 hourly averages over 2 in filtered air. Removing PAs with high CH1 offsets in filtered air reduces uncertainty and improves precision, particularly in cleaner ambient air.

### 2.2.8 PMS CH1 unresponsive to $CO_2$ and Suva[®]

Filtered air, $CO_2$, and Suva[®] (DuPont™ Suva[®] 134a refrigerant) are often used to calibrate integrating nephelometers (Anderson et al., 1996). The Rayleigh scattering coefficients of filtered air, $CO_2$, and Suva at 657 nm and at STP (0 °C and 1013.25 hPa) are 5.5, 13.3, and 46.2 Mm$^{-1}$, respectively. We found that the PMS was unresponsive to 100% $CO_2$ (Fig. S11) and Suva. The CH1 for each gas was the same as filtered air. These results indicate that the PMS signal processing zeroes out a constant scattering signal and cannot be used to measure the scattering coefficient of gases that are commonly used in calibrating nephelometers. Furthermore, the method used by the PMS to subtract light scattering by air molecules in the sampling volume is unknown.

### 2.2.9 PMS CH1 and CH1avg precision

The PMS CH1 precision was measured by collocating ten PA monitors on the roof of the NOAA building in Boulder, Colorado, between 22 January 2021 and 1 February 2021. These monitors

were not checked with filtered air before deployment. It was found that two of the PMS sensors
had large offsets and two had moderate offsets at low CH1 values. One PMS sensor was found to
produce errant data and was removed from the analysis, resulting in valid data from 19 CH1A
and CH1B sensors in the ten PAs.
The precisions for the hourly data from the CH1A and CH1B sensors and their average
(CH1avg) were estimated as the coefficient of variation for each of the 19 CH1A and CH1B
values and the 9 CH1avg values for each hour, which are plotted against the average CH1 values
in Fig. 2.  As shown, above CH1 values of 500, the precision is relatively constant with an
average of 8% and 4.8% for CH1A-CH1B and CH1avg, respectively.  Below CH1 values of 500,
the uncertainties increase rapidly with decreasing CH1 values.
There are two mechanisms that may contribute to the rapid uncertainty increase for CH1 < 100.
First, it is likely that some of the increased uncertainty in CH1 below values of 100 is inherent to
sampling low concentrations, as is the case for any instrument.  Second, the geometry of the laser
sensing volume in the PMS can contribute to uncertainty in the CH1 at low concentrations,
specifically if particles are not distributed uniformly within the laser beam.
The data in Fig. 2 can be modeled by the sum of squares of an additive ($Un_{add}$) and multiplicative
uncertainty ($Un_{mult}$) (Currie, 1968; Hyslop and White, 2008; JCGM100:GUM, 2008):
$$Uncertainty \;=\; \sqrt{Un_{add}^2 + Un_{mult}^2 * CH1} \tag{1}$$
Equation 1 was fitted to the precision data in Fig. 2 where the $Un_{mult}$ was set to the average
precision at high CH1 values, and $Un_{add}$ was set to 28 and 19 for the A and B sensors and
CH1avg, respectively, to fit the highest variances (Table S2).  The $Un_{add}$ is the precision of CH1
as CH1 approaches zero and is assumed to be equivalent to the uncertainty in values below the
instrument minimum detection limit (MDL) or that of blanks (Currie, 1968), which were 0.08
and 0.048 for the A and B sensors and CH1avg, respectively.  The coefficient of determination in
the model fit for both sets of data was $r^2 = 0.96$.  Defining the MDL as the 99% confidence
interval of the $Un_{add}$ (Code of Federal Regulations, 40 CFR 136, https://ecfr.io/Title-40/Part-
136), MDLs for the individual CH1 sensors and CH1avg were 65 and 44, respectively.
As shown in Sect. S3, the $Un_{mult}$ and $Un_{add}$ are highly dependent on the systematic biases
between the individual CH1 sensors and CH1avg and the four CH1 sensors with data offsets as
the CH1 approaches zero (Fig. S12).  Removing these four sensors and normalizing the data for
each CH1 sensor by its average reduced the $Un_{add}$ and $Un_{mult}$ to 9% and 3%, respectively, for the
CH1 sensors and 6% and 1.9%, respectively, for the CH1avg data.  These results correspond to
an MDL of 21 and 14 for the normalized CH1 sensor and CH1avg data, respectively.  Based on
these results, an "off the shelf" PA will have a CH1avg MDL of about 44 and precision of less
than 4.3%, but the careful selection of a PA without an offset and that has relatively low noise
will have an MDL of 14 and precision of less than 1.9%.
**2.3 Overview of cell-direct and cell-reciprocal nephelometers**
The integrating nephelometer was invented during World War II (Beuttell and Brewer, 1949). It
provides a direct measure of aerosol light scattering integrated over a large angular range, the
"aerosol light scattering coefficient". This measure requires no assumptions about aerosol
composition, size distribution, refractive index, or shape. The most common nephelometer
configurations are the "cell-direct" and "cell-reciprocal". Figure 3 presents schematics of the two
types of nephelometers. The geometrical relationship between the laser and the photodetector in
the PMS resembles a cell-reciprocal nephelometer (Fig. 3b).
Middleton (1952) was the first to show that the cell-direct nephelometer with a Lambertian
(cosine-adjusted diffuser) light source directly measures the aerosol light scattering coefficient.
Anderson et al. (1996), following the derivation in Butcher and Charlson (1972), added
geometrical diagrams to make Middleton's derivation much clearer. Mulholland and Bryner
(1994) proved that the cell-reciprocal nephelometer with a Lambertian diffuser followed by a
photodiode placed at the center of the cell-reciprocal nephelometer also directly measures the
aerosol scattering coefficient. This put both the cell-direct and cell-reciprocal nephelometers on
equal theoretical footing.
There are a number of cell-direct nephelometers in use today. They include the TSI 3563 (St.
Paul, MN, USA; Anderson et al., 1996), the Ecotech Aurora Models 3000 and 4000 (Knoxfield,
Australia; Müller et al., 2011), the Radiance Research M903 (Seattle, WA, USA; Heintzenberg
et al., 2006), and the Optec NG-2 (Lowell, MI, USA; Molenar, 1997). In contrast, cell-
reciprocal nephelometers have more limited commercial availability. The photoacoustic
extinctiometer (PAX; Droplet Measurement Technologies, Inc., Longmont, CO, USA) and the
three-wavelength photoacoustic soot spectrometer (PASS-3) use a cell-reciprocal nephelometer
to measure the aerosol light scattering coefficient (Arnott et al., 2006). A cosine corrector
followed by a photomultiplier tube is placed at the center of the cell-reciprocal nephelometer
(Abu-Rahmah et al., 2006; Nakayama et al., 2015).
A "perfect nephelometer" is one in which the nephelometer is able to see the scattered light over
the entire angular range from 0 to 180 degrees. In practice, this cannot be achieved for the cell-
direct and cell-reciprocal nephelometers. Both the forward and backward scattering angles are
truncated. For example, the TSI 3563 nephelometer has measured angular truncation below
about 7 degrees in the forward direction and above 170 degrees in the backward direction
(Anderson et al., 1996; Heintzenberg and Charlson, 1996). For the PASS-3, Nakayama et al.
(2015) found that both the large effective truncation angle (21 degrees) as well as the
perpendicular polarization of the 532 nm laser relative to the scattering plane contribute to the
large particle size dependence of measured scattering. Light scattering from ammonium sulfate
particles of 0.71 μm diameter was reduced by 50% relative to a perfect nephelometer. Angular
truncation generally results in nephelometers underestimating the contribution of particles larger
than approximately 1 μm diameter to the scattering coefficient, although corrections have been
developed to account for angular nonidealities (e.g., Anderson and Ogren, 1998; Müller et al.,
407 2011).

## 3. A physical-optical model of the PMS5003

To gain insight into how the PMS responds to ambient aerosol properties, a model was
developed to estimate the intensity of scattered light impinging on the PMS photodiode. The
primary purpose of the model was to predict how the PMS performance compares to other
instruments designed to measure the aerosol scattering coefficient, such as integrating
nephelometers. The model makes simplifying assumptions about the laser that allow the
application of Mie theory to the light scattered from particles in the laser.  Details of the model
are presented in the Appendix.
The equation describing the intensity of light scattered from a particle in the laser is (Middleton,
1952; Anderson et al., 1996)
$I(\theta) = F_{dv}\, \beta_p(\theta)\, dv$                                                   (2)
where $I(\theta)$ is the intensity of light at angle $\theta$ scattered from a particle in the volume element dv
(with units of W sr$^{-1}$); $\beta_p(\theta)$ is the volume scattering function (m$^{-1}$ sr$^{-1}$); $F_{dv}$ is the incident laser
flux density (W m$^{-2}$) impinging on the volume element dv; and dv is the volume element within
the laser.
The volume scattering function for a single particle in the laser beam is a function of aerosol
diameter $D_p$, complex refractive index m, laser wavelength $\lambda$, and scattering angle $\theta$:
$\beta_p(\theta) = (\lambda/2\pi)^2\, (1/dv)\, |S_1(m, \lambda, \theta, D_p)|^2$                              (3)
where $|S_1(m, \lambda, \theta, D_p)|^2$ is the Mie scattering intensity function for laser light polarized parallel to
the photodiode surface and perpendicular to the plane of incidence (Bohren and Huffman, 1983).
The scattered light intensity from a single particle in the laser beam to a narrow strip across the
middle of the photodiode and from all positions in the scattering volume is integrated to predict
the total power received by the photodiode as a function of particle diameter $D_p$ and refractive
index m:
$P(m,D_p) = K \int_{x=0}^{x=10mm} \int_{\theta 1(x)}^{\theta 2(x)} |S_1(m,\theta,D_p)|^2 \sin(\theta)\, d\theta\, dx.$          (4)
Due to the PMS geometry, the upper and lower angular scattering limits for $\theta$ depend on the
location, x, of a particle in the laser. Details are provided in the Appendix. This approach can be
used to estimate the amount of scattered energy detected from mixtures of particles of varying
diameters and indices of refraction, as shown in Eq. (5):
$P = K \int_{Dp} \int_{x=0}^{x=10mm} \int_{\theta 1(x)}^{\theta 2(x)} |S_1(m,\theta,D_p)|^2 \sin(\theta)\, N(D_p, m)\, d\theta\, dx\, dD_p.$          (5)
**3.1 Model predictions - Deviation from a perfect cosine response**
As discussed above, the PMS has a photodetector that is about 1.8 mm below the laser, resulting
in forward scattering and backscattering truncation angles of 18 and 166 degrees, respectively.
Furthermore, the photodetector is recessed 0.46 mm below the scattering chamber base.
Equation 4 is used to explore the deviation from a perfect cosine response resulting from the
truncated scattering volume and recessed detector. It is shown in Fig. 4.  For these calculations,
$S_1(m,\theta,D_p)$ is set equal to 1, which corresponds to isotropic scattering or a volume scattering
function that is constant over all scattering angles.  It is assumed that the detector has a
Lambertian response, i.e., the light detected is independent of the direction of the incident
energy, which results in a detector cosine response.  Figure 4 shows a perfect cosine response in
yellow, while the red line shows the deviation from a perfect cosine response due to angular
truncation. The blue line shows the effect of both angular truncation and an inset detector that is
0.46 mm below the chamber base. All curves have been normalized to one at 90 degrees.
**3.2 Model predictions - Intensity versus position on the detector**
Figure 5 provides an example of the energy distribution on the photodiode as a function of
position in the laser and on the diode resulting from scattering from particles represented by a
lognormally distributed aerosol volume size distribution with a volume mean diameter of 0.33
μm and geometric standard deviation of 1.7. Figure 5 shows model predictions of the relative
intensity of scattered light, where the values are proportional to energy flux impinging on the
detector.
The masking resulting from a recessed detector truncates the scattering both in the most forward
and most backward scattering angles. This masking is shown as the triangular area
corresponding to distance down the laser and detector of 0.0–2.5 mm and 0.0–0.78 mm,
respectively, for the forward scattering angles and 5.6–10 mm and 1.44–3.0 mm, respectively,
for backscattering. Because the laser is parallel to the photodetector, which is assumed to have a
$\cos(90-\theta)$ response, the maximum energy scattered to the detector is approximately at $\theta = 90$
degrees. However, more energy is scattered to the detector for scattering angles less than 90
degrees, which corresponds to forward scattering, and very little energy is detected by the
photodiode for particles in the laser that are greater than about 8 mm down the laser beam, even
though the detector is exposed to particles in the laser that are 10 mm away from the laser exit
hole. These distances down the laser correspond to backscattering. The total energy detected by
the photodiode is the sum or integral across both the detector surface and position in the laser
and corresponds to the volume under the curve depicted in Fig. 5.
**3.3 Model predictions - Predicted photodiode response as a function of particle diameter**
The PMS differs from a perfect nephelometer in at least five important ways:
1. The laser is polarized, whereas the nephelometer light source is unpolarized.
2. The laser beam profile is not a simple plane wave, but complex in shape. The laser beam
475        profile evolves significantly as it is focused over the photodiode.
3. The photodiode likely does not have a perfect cosine response.
4. The PMS geometry limits the photodiode to receiving scattered light between
478        approximately 18 and 166 degrees, whereas a perfect nephelometer measures all energy
479        scattered between 0 and 180 degrees.
5. The unknown PMS signal processing removes the light scattering signal from $CO_2$, Suva,
481        and filtered air. These gases are used to calibrate nephelometers but cannot be used to
482        calibrate the PMS.

The effects of these differences can be seen in Fig. 6, which shows predicted photodiode
response as a function of particle diameter. The perfect nephelometer response is in blue, and
the PMS response is in yellow. The red line predicts PMS response if the laser were not
polarized. Relative intensities have been normalized to an ideal nephelometer measurement of a
0.1 μm diameter particle, which is akin to adjusting the laser power such that the scattered power
at a diameter equal to 0.1 μm is the same for all configurations. Scattering as a function of
particle diameter is nearly the same for all three configurations from 0.1 μm to about 0.3 μm. At
about 0.8 to 1.0 µm, the response of a PMS with an unpolarized laser is about half that of an
ideal nephelometer, and the use of a polarized laser reduces its response to about 30% to that of
an ideal nephelometer.  For particles above 2 µm in diameter, the PMS response compared to an
ideal nephelometer is decreased by about 75%.  Additionally, the PMS manual (Zhou, 2016)
quotes a lower detection limit diameter of 0.3 µm.  The model predicts that particles smaller than
0.3 µm in diameter would be detected by the PMS, in direct proportion to their contribution to
the scattering coefficient.
These differences in geometry and optics from an ideal nephelometer are further highlighted in
Fig. 7. To highlight the effect of polarization, the blue line shows the ratio of an ideal
nephelometer with a laser light source that is perpendicularly polarized to an ideal nephelometer
with an unpolarized light source while the red line shows just the effect of PMS geometry
relative to an ideal nephelometer.  The yellow line shows the effects that polarization and PMS
geometry have on the measured scattering signal.  Again, all hypothetical instrument responses
have been normalized to a particle diameter of 0.1 µm. Relative to scattering for a 0.1 µm
particle, the polarization alone reduces the scattering signal of an ideal nephelometer by 40% for
particles with diameters in the 0.8–1.5 µm size range.  The additional effect of PMS scattering
geometry reduces the scattering signal at 0.8–1.0 µm by about another 30% relative to an ideal
nephelometer.
As noted in Sect. 2.2.3, the specifications of the BPW34 are used to estimate the likely properties
of the detector in the PMS.  Our model assumes two ideal properties of the photodiode. The first
is area uniformity - that a photon impinging any part of the photodiode would generate the same
current as the same photon impinging on another part of the photodiode. The second ideal
assumption is that the dependence of the photodiode current on the light intensity is very linear
over 4 or more orders of magnitude.  If these assumptions do not hold, then the yellow curve in
Fig. 7 will change.
The variance in the PMS physical and optical geometry and errors in the measurements are not
known but likely small.  To evaluate the sensitivity of the modeled PA scattering to errors in
these measurements, the model was exercised with large deviations of ±25% and ±50% in these
inputs.  As shown in Table S3, the errors tend to increase with particle size.  The modeled PA
scattering to a perfect nephelometer is most sensitive to errors in the distance from the laser to
the photodiode. For particle diameters of 0.5 µm, +25% and +50% changes in this distance
resulted in maximum differences of 10% and 20%, respectively.  Based on these results and the
fact that the errors in the physical dimensions are less than 25%, these errors are thought to have
a small contribution to the overall modeled PA scattering error and were not directly accounted
for in the analysis.  This analysis does not attempt to account for the possibility that the laser
beam profile is not a simple plane wave or that the laser beam profile may evolve significantly as
it is focused over the photodiode, and the standard plane wave Mie calculations would no longer
apply.

### 528    3.4 Model predictions – Differentiating by particle size

The irradiance received by the PMS photodiode from a particle of a given diameter and
refractive index depends on the particle's location in the laser beam. The model predicts that
particles of different sizes may contribute the same irradiance to the photodiode, depending on
their location in the beam, or conversely, light scattered by a particle of a given size can vary by
more than an order of magnitude.
As an example, the model predicts that all of the particles in Fig. 8 contribute the same irradiance
to the PMS photodiode.  The smaller particles contribute the same irradiance by scattering in the
more effective forward scattering regime. The larger particles contribute the same irradiance by
scattering in the less effective backscattering regime.  The photodiode and its associated
electronics would not be able to differentiate between them. As a result, the model predicts that
the values reported in the six PA-PMS particle size channels from >0.3 μm to >10 μm cannot
correctly represent the aerosol size distribution.

## 4. Experimental – Field studies

Field experiments were conducted at two of the NFAN aerosol monitoring stations: the Mauna
Loa Baseline Observatory in Hawaii and the Table Mountain Test Facility in Colorado.  Both
sites have large suites of aerosol instrumentation and daily access for scientists and technicians to
inspect, calibrate, and maintain the instruments. These sites also have integrating nephelometers
(TSI 3563, St. Paul, MN, USA) against which to evaluate the PA monitors.

### 4.1 Description of Mauna Loa site

The Mauna Loa Baseline Observatory (MLO) is located on the north flank of the Mauna Loa
volcano, on the Big Island of Hawaii (19.536 ºN, 155.576 ºW, 3397 m asl). The observatory is a
premier atmospheric research facility that has been continuously monitoring and collecting data
on global background conditions and atmospheric change since the 1950s
(https://www.esrl.noaa.gov/gmd/obop/mlo/). Continuous aerosol measurements at MLO began in
the mid-1970s with the installation of condensation particle counters and an integrating
nephelometer (Bodhaine and Mendonca, 1974; Bodhaine et al., 1981). MLO lies above the
strong marine temperature inversion layer present in the region, which separates the more-
polluted lower portions of the island atmosphere in the marine boundary layer from the much
cleaner free troposphere. MLO experiences a diurnal wind pattern (Ryan, 1997) that is strongly
influenced by the daily heating and nighttime cooling of the dark volcanic lava rock that makes
up the mountain. This "radiation wind" brings air up from lower elevations during the daytime,
when atmospheric measurements reflect the local mountain environment. In contrast, during the
nighttime, downslope winds develop, and the measurements at MLO are typically dominated by
clean, free-tropospheric conditions (Chambers et al., 2013). At these times, the aerosol
measurements at MLO often reflect some of the cleanest conditions at any station in the northern
hemisphere. It has long been known, however, that episodic long-range transport of Asian
pollution and dust aerosols occurs, most frequently in the springtime (Shaw, 1980; Miller, 1981;
Harris and Kahl, 1990), and these aerosol events can influence both the daytime and nighttime
measurements at MLO.  Consequently, the aerosol levels at MLO vary over a large range, from
extremely low to at times mildly elevated.  Here we use observations from the MLO integrating
nephelometer to evaluate the PMS sensor.

### 4.2 Description of Boulder Table Mountain site

The Table Mountain Test Facility (BOS) is a large restricted-access federal complex located 14
km north of Boulder, Colorado (40.125 ºN, 105.237 ºW, 1689 m asl). NOAA conducts
atmospheric research at this site, and in addition to its NFAN station, it is one of the Global
Monitoring Laboratory's seven U.S. Surface Radiation Network (SURFRAD) sites
(https://www.esrl.noaa.gov/gmd/grad/surfrad/tablemt.html). Many instruments for measuring
surface and column aerosol properties are maintained at this location and used for long-term
monitoring of the atmosphere.
The BOS site lies just east of the Front Range foothills of the Rocky Mountains and is typical of
a semi-arid, high plains environment. It is a high mesa of predominantly grassland with some
desert scrub vegetation. The location is well suited for sampling of wildfire smoke plumes during
fire season in the western United States (summer and autumn), dust events at any time of the
year, and occasional urban pollution episodes. The NFAN station at BOS
(https://www.esrl.noaa.gov/gmd/aero/net/bos.html) was completed in September 2019. BOS
operates an integrating nephelometer and a differential mobility particle spectrometer (DMPS).
Both provided useful data for evaluating some of the predictions from the physical-optical model
we developed for the PMS sensor.

### 4.3 PA monitors

PA-PMS monitors were installed on the aerosol towers at the MLO and BOS stations, just below
the main aerosol inlets. MLO had two PA-PMS monitors, one gently heated and one unheated,
whereas BOS had one gently heated PA-PMS monitor. Prior to deployment, the monitors were
tested in a filtered air chamber for 4 hours to ensure that the 1 h average CH1 values were less
than 1 when no particles were present. One of the PMS sensors in the unheated MLO PA had 1 h
average CH1 values of 27 when no particles were present.  The heated monitors were wrapped
with heating tape and powered by small DC power supplies. All the monitors were covered with
stainless steel flashing 5 cm below the bottom to prevent rain and snow from entering the inlet
(Fig. S13).
The PA-PMS monitors were warmed in an effort to reduce the sample RH to be closer to that of
the nephelometer, which is unavoidably heated to above ambient temperatures by the warmth of
the laboratory and by the nephelometer's halogen lamp. Because of this warming, the RH inside
the nephelometers rarely exceeded 40%. Both MLO and BOS are low-RH environments under
normal conditions, although occasionally moist air masses are encountered.  The heating of the
monitors increased the sample temperatures by 5–8 ºC, which helped to lower the sample RH.
While the PA heating was not controlled to achieve an RH match with the nephelometer, it
brought the sample RH of the two measurements closer together. The gentle warming of the
heated PA to only a few degrees above ambient is unlikely to cause the PVC to off-gas or melt.
Heating from direct sunlight may have had a larger impact.
Due to internet protocols at both sites, PA's wireless data transmission feature was not used, and
the data were stored on the internal micro SD card. At approximately 1 month intervals, the data
were downloaded from the micro SD cards, and the PAs were returned to service. Outputs from
the two PMS sensors were then compared at these intervals to determine if the PAs were still
functioning properly. In this study, the 80 s or 2 min averages were used to create 1 h averages to
compare the PA observations to those of the nephelometer and the DMPS.

### 4.4 Integrating nephelometer

The integrating nephelometer (TSI Inc., model 3563) measures the aerosol light scattering coefficient at three wavelengths (450, 550, and 700 nm). At both sites, the sample flow path is switched every 6 min between 1 and 10 μm aerodynamic diameter, multijet, Berner-type impactors. Here, the scattering coefficients at 550 nm for both the $PM_1$ and $PM_{10}$ size fractions are used for comparison with the PMS measurements. These are referred to as $b_{sp1}$ and $b_{sp10}$, respectively.

There are two quality checks of the nephelometer operation made in the field. First, the nephelometer automatically samples filtered air once per hour. This provides a record of the stability of the instrument background measurement. Second, the nephelometer calibration is manually checked on a monthly basis using $CO_2$ and filtered air (Anderson et al., 1996). The 1 h average $b_{sp1}$ in filtered air is 0.01 $Mm^{-1}$ with a standard deviation of 0.12 $Mm^{-1}$, based on 125 hours of sampling filtered air.

The nephelometer measurements were corrected for angular truncation (Anderson and Ogren, 1998) and reported at STP. Weekly data review provides quality assurance of the nephelometer data. Scattering coefficient data were averaged to 1 min resolution for logging and were further averaged to hourly resolution for comparison with the PA data. The 1 h average $b_{sp1}$ uncertainties of the nephelometer measurements are ~0.13 $Mm^{-1}$ for scattering coefficients less than 1.0 $Mm^{-1}$ and ~10% for scattering coefficients greater than 1 $Mm^{-1}$ (Sherman et al., 2015).

### 4.5 Differential mobility particle spectrometer (DMPS)

The DMPS was provided by the Institute for Atmospheric and Earth System Research, University of Helsinki, Finland. It was checked and calibrated by the World Calibration Centre for Aerosol Physics (WCCAP) at Leibniz Institute for Tropospheric Research (IfT), Leipzig, Germany, just prior to deployment at NOAA's Table Mountain site. After shipment from IfT to NOAA, the DMPS was again checked by aerosolizing polystyrene latex spheres and confirming that the peaks occurred in the correct size bins. The DMPS was housed inside the same building as the nephelometer at BOS and sampled aerosols through the same inlet, although the DMPS flow did not pass through the aerosol impactors.

The DMPS provides 40 channels of particle concentration versus size, ranging from mobility diameters of 0.01 μm to 0.8 μm. The 0.1 μm to 0.8 μm channels of the DMPS were used to calculate hourly average fine aerosol scattering coefficient distributions and the total fine aerosol scattering coefficient, assuming spherical particles (Mie theory) with a refractive index of 1.53+0.017i. The hourly average, DMPS-calculated fine aerosol scattering coefficients were compared to the nephelometer-measured fine aerosol scattering coefficients to check operational consistency (Fig. S14). No operational changes were made to the DMPS during this field study. This study did not measure coarse aerosol size distributions. The DMPS hourly average fine aerosol scattering coefficient distributions were used with the PMS physical-optical model to predict total 1 h average scattered irradiance on the photodiode.

## 5. Results

This section describes our evaluation of the PA-PMS using field data from MLO and BOS. First, we provide an overview of the observational data. We then assess how well the model described

in Sect. 3 is able to represent the observed data and show consistency with results previously
reported in the literature. Next, we present results showing the potential of the PA-PMS to
perform as a nephelometer. Finally, we note how the size information output by the PA is not
correct due to the PA's primary measurement being a scattering measurement. For the results
presented below, data from the PA-PMS, nephelometer, and DMPS were averaged to hourly
frequency and merged prior to analysis.

## 5.1 Field data overview

Heated PA monitors were deployed at the MLO and BOS observatories for 15 and 11 months,
respectively (Table 1). At both sites weather had no impact on the operation of the PA
instrument, and downtime only occurred during data downloading.
These two deployments provide an excellent dataset for assessing PA performance in both a
clean location (MLO) and in an environment with more elevated particle concentration (BOS).
As shown in Table 2, during the field study at MLO, the median CH1 was 26.7. The median $b_{sp1}$
was 0.76 Mm$^{-1}$ at 550 nm, which is approximately 10% of Rayleigh scattering at the MLO
altitude. The reported PM$_{2.5}$ mass concentration from the PA was zero for most of the MLO
deployment. The CH1 and $b_{sp1}$ are adjusted to STP in Table 2. The air quality at BOS was less
pristine than at MLO and is more representative of nonurban continental air quality. The very
high maximum CH1 and $b_{sp1}$ at BOS reported in Table 2 occurred during smoke events in the
summer and autumn of 2020. One of the BOS PMS sensors experienced approximately 10%
degradation in sensitivity after one year in the field (Fig. S15).

## 5.2 Relationship between model predictions and field data

The PMS sensor is described by the manufacturer as a particle counter that measures particles
between 0.3 μm and 10 μm in six size bins. Based on the theoretical characterization of the PMS
sensor described in Sect. 3, the sensor is more akin to a polarized, reciprocal integrating
nephelometer than a particle counter. Below, the field data and theoretical model are used to
demonstrate that the raw PMS CH1 sensor signal is an integrated scattering measurement that is
sensitive to particles smaller than 0.3 μm but relatively insensitive to particles larger than 1.0
μm.

### 5.2.1 Predicted photodiode irradiance versus CH1 field data at BOS

Our model described in Sect. 3 and the Appendix predicts a value proportional to the scattered
irradiance impinging on the PMS photodiode as a function of particle diameter and
concentration. This was done using the DMPS size distribution data from BOS. The modeled
PMS photodiode output is plotted against the PMS CH1 output (Fig. 9). The predicted
photodiode output is linearly correlated with the ordinary least squares (OLS) regression ($r^2$ =
0.90, normalized root mean square error (NRMSE) ~25%) with CH1 over 4 orders of magnitude.
The RMSE contains contributions of errors from the model-predicted radiant power, the
measured SMPS data the model is based on, as well as in the CH1 measurements. This strong
correlation and low RMSE is convincing evidence that the model and SMPS data describe the
PMS response quite well.
The linear relationship between CH1 and modeled photodiode response suggests the likelihood
that the CH1 output is directly related to what the photodiode is sensing (i.e., scattering from all
particles in the scattering volume).  The PA-PMS reported values, such as concentrations of
particle numbers in various size ranges or PM concentrations, are quantities derived from the
scattering signal and the use of an undescribed algorithm.

### 5.2.2 Predicted aerosol size truncation versus published laboratory data

The PMS physical-optical model described in Sect. 3 predicts that if CH1 is proportional to the
photodiode power, then its signal will be reduced relative to a perfect nephelometer. Thus, the
ratio $CH1/b_{sp}$ should decrease as median scattering diameter increases. To test this prediction,
data were obtained from published laboratory studies evaluating the PMS against aerosols of
varying composition and size distribution reported by Tryner et al. (2020) and He et al. (2020).
These reported aerosol size distributions were used here to calculate the aerosol scattering
coefficient distributions from 0.1 to 10 μm for the various aerosols and refractive indices at a
wavelength of 657 nm.  The median scattering diameter (MSD) was calculated for each test. The
MSD is the aerosol diameter at which approximately half of the light scattering coefficient is due
to particles smaller than the MSD and the other half to particles larger than the MSD. The MSD
was then compared to the ratio of the measured CH1 and $b_{sp10}$ values, i.e., $CH1avg/b_{sp10}$, for each
of the tests reported in Tryner et al. (2020) and He et al. (2020).  Figure 10 summarizes the
results for $CH1/b_{sp10}$ versus MSD.
The controlled laboratory results are in general agreement with the PMS physical-optical model,
showing substantial reduction in $CH1avg/b_{sp10}$ as a function of increasing particle diameter from
0.2 to 1μm.  The laboratory results show an even greater reduction in $CH1avg/b_{sp10}$ than the
model predicts at diameters larger than 1 μm. This suggests the possibility of supermicron
aerosol loss before laser detection, perhaps due to aspiration as discussed in Sect. 2.2.1.

### 5.2.3 $CH1avg/b_{sp1}$ as a function of median scattering diameter

Although ambient aerosols may vary considerably in composition and morphology and cannot be
as simply characterized as laboratory aerosols, it is instructive to evaluate if PMS angular
truncation can be observed using field data. The DMPS data from BOS were used to calculate
hourly average aerosol scattering coefficient distributions for diameters between 0.1 μm and 0.8
μm. A wavelength of 657 nm and a particle refractive index of 1.53+0.0i were used for the
calculations.  The median scattering diameter was calculated for each hour. The MSD was then
compared to the ratio of the measured CH1 and $b_{sp1}$ values, i.e., $CH1avg/b_{sp1}$, for each of these
hours. The results are shown in Fig. 11 as a box and whisker plot of the $CH1avg/b_{sp1}$ values
found in each MSD bin.  The center MSD value for each bin is based on a logarithmic scale of
MSD values where the upper and lower bin values are selected as $MSD_i+MSD_{i+1}/2$ and $MSD_i-$
$MSD_{i-1}/2$ and i refers to the $i^{th}$ bin. The thin black horizontal lines correspond to the number of
observations in each bin and the scale is shown on the right hand axis. There are less than 20
values in the 0.22 μm, 0.63 μm, 0.71 μm, and 0.79 μm bins. Approximately 67% of the MSDs
observed at BOS were between 0.29 μm and 0.36 μm, and 98% of MSDs were between 0.26 μm
and 0.46 μm. The overall average $CH1avg/b_{sp1}$ ratio, based on 6777 observations, is 65 Mm.
Figure 11 is consistent with the PMS physical-optical model.  The highest $CH1avg/b_{sp1}$ ratios
tend to occur for aerosols with the lowest MSD and decrease as MSD increases.  Additionally,
the results show, as suggested above, that the PMS can detect particles below 0.3 μm in diameter
in proportion to their contribution to the scattering coefficient.

### 5.2.4. Estimating the scattering coefficient minimum detection limit of the PA-PMS

The precision analysis in Sect. 2 indicates that the PA monitors used in this study estimated 1 h average CH1 and CH1avg MDLs of approximately 21 and 14, respectively. The estimated 1 h average MDL $b_{sp1}$ of the TSI 3563 nephelometer is approximately 0.20 Mm$^{-1}$, based on filtered air tests. Further analysis of the relationship between CH1 and $b_{sp1}$ at low levels was performed by plotting the ratio, CH1avg/$b_{sp1}$, for the combined MLO and BOS dataset, as a function of $b_{sp1}$. This relationship is shown graphically in Fig. 12. The data values were first averaged over 6 hours because hourly $b_{sp1}$ values near zero included many small negative $b_{sp1}$ values due to the very clean conditions occasionally observed at MLO. The averaging eliminated all but five negative $b_{sp1}$ values, which were removed from the dataset. The CH1avg/$b_{sp1}$ and $b_{sp1}$ values were further averaged over six data points after sorting the data on $b_{sp1}$ levels to more clearly show the relationship between CH1avg and $b_{sp1}$. At $b_{sp1} > 5$ Mm$^{-1}$, the CH1avg/$b_{sp1}$ ratio is relatively constant at 67 Mm, the yellow line in Fig. 12. The yellow line is the slope of CH1avg versus $b_{sp1}$ at $b_{sp1}$ values greater than 5 Mm$^{-1}$. The CH1avg/$b_{sp1}$ ratio systematically decreases from its highest values to about 35 Mm, the slope of CH1avg versus $b_{sp1}$ at $b_{sp1} = 0.4$ Mm$^{-1}$. For $b_{sp1} < 0.4$ Mm$^{-1}$ the CH1avg/$b_{sp1}$ ratio then increases significantly as $b_{sp1}$ decreases, consistent with CH1avg values staying approximately constant below 0.4 Mm$^{-1}$. Both the CH1avg and $b_{sp1}$ are below MDL for $b_{sp1} < 0.2$ Mm$^{-1}$. A CH1avg/$b_{sp1}$ ratio of approximately 35 Mm at $b_{sp1} = 0.4$ Mm$^{-1}$ and a CH1avg value of about 14±5 is consistent with the estimated CH1avg MDL of 14.

Based on these results, the 1 h average CH1 sensor MDL for hourly data in units of scattering is approximately 0.4 Mm$^{-1}$ at MLO. Laboratory tests challenging the PAs with known low-level, spiked aerosol concentrations and defined size distributions are needed to further refine the estimated MDL.

### 5.2.5 Evaluating the use of the PA-PMS as an integrating nephelometer

The MLO and BOS hourly average CH1avg are plotted against $b_{sp1}$, measured at 550 nm, in Fig. 13. Also shown in Fig. 13 is an OLS regression line with the intercept set equal to zero using the BOS and MLO combined dataset but with values associated with $b_{sp1}$ less than 0.4 Mm$^{-1}$ and greater than 500 Mm$^{-1}$ removed. Results of the regression for the combined datasets as well as for the individual BOS and MLO datasets are presented in Table 3. There is good agreement for both datasets (Table 3) with an $r^2$ of 0.97 and 0.85 for the BOS and MLO datasets, respectively, and 0.97 for the combined datasets. The relationship deviates somewhat from linear with increasing slopes and scatter at lower values of atmospheric scattering coefficient, particularly for the MLO data. The slopes (in Mm$^{-1}$) for all data, MLO, and BOS, are 0.015±2.07×10$^{-5}$, 0.017±5.72×10$^{-5}$, and 0.015±2.68×10$^{-5}$, respectively. In the following analysis, a PA-derived atmospheric scattering ($b_{sp1,PA}$, Mm$^{-1}$) for both MLO and BOS is estimated using $b_{sp1,CH1} = 0.015$ × CH1avg at a wavelength of 550 nm. The best-fit value of 0.015 Mm$^{-1}$ corresponds to the yellow horizontal line in Fig. 12 of 67.0 (1/0.015) and corresponds to a median scattering diameter of about 0.33 μm (Fig. 11).

Figure S16 shows that the submicron aerosol scattering coefficients at 550 nm and 700 nm are highly correlated, with the 700 nm scattering coefficient averaging 52% of the 550 nm scattering coefficient. This results in $b_{sp1,CH1} = 0.0078$ × CH1avg at a wavelength of 700 nm.

As discussed above, the regression coefficient between $b_{sp1}$ and CH1 for the combined dataset of
0.015 $Mm^{-1}$ is used to estimate the $b_{sp1,PA}$ derived from the CH1 channel. The data for each
dataset and the combined dataset were binned into ten bins based on measured $b_{sp1}$ levels that
ranged from 0.4 $Mm^{-1}$ to 500 $Mm^{-1}$. Values of $b_{sp1}$ above 500 $Mm^{-1}$ were removed from the
dataset. For each bin the NRMSE between $b_{sp1,PA}$ and measured $b_{sp1}$ was calculated. The
NRMSE values as a function of the $b_{sp1}$ bins are plotted in Fig. 14 for the combined dataset
represented as the gray bars and BOS and MLO represented by blue and orange bars,
respectively.
For $b_{sp1}$ levels less than 0.8 $Mm^{-1}$, the NRMSE is 45–55%, and for $b_{sp1}$ levels greater than 10
$Mm^{-1}$, the NRMSE is about 25% or less. For $b_{sp1}$ levels greater than 60 $Mm^{-1}$, the NRMSE
approaches 15%.
As discussed in Sect. 2.2.9, the uncertainty for high CH1avg values is small (1.9% to 4.8%). The
precision of the TSI 3563 nephelometer is also similarly high, and together they account for
about 10% NRMSE at high $b_{sp1}$ values.
The overall normalized error is likely due to a variety of sources, primarily the variability in the
CH1 values due to using a polarized light source and truncation errors due to the geometry of the
PA-PMS sensors. Also, the variability in aerosol characteristics such as size distribution,
refractive index, and shape may be important. At extremely low levels, uncertainty may also be
due to a nonuniform distribution of particles in the PMS laser beam.
There are two reasons why the PA-PMS MDL and RMSE values reported in our study are
surprisingly low. The TSI 3563 nephelometer has an extremely low detection limit of 0.20 $Mm^{-1}$,
which is approximately 1% of Rayleigh scattering. Second, the PA-PMS has very low noise at
zero aerosol concentration. If the PA in our study had been collocated with a nephelometer that
was not as sensitive as the TSI 3563 in a location having an average fine aerosol coefficient of,
say, 30 $Mm^{-1}$, then the PA 1 h average MDL could have been significantly higher than the 0.4
$Mm^{-1}$ we obtained in our study.
**5.2.6 PA-PMS size distributions**
The aerosol number concentrations from the six PMS size channels are unrealistic. The BOS
field data showed that the concentration of particles larger than 0.3 μm diameter calculated from
the DMPS averaged 10 times higher than CH1 (Fig. S17). The other PMS size channels are so
highly correlated with CH1 that they provide no additional information (Table S4). Furthermore,
it appears that the PMS creates an approximately invariant normalized aerosol number
distribution across a wide range of sites (Table S5, Fig. S18). Although the overall CH1
concentration can vary over 6 orders of magnitude (column 3 in Table S5), the shape of the PMS
size distribution remains fairly constant.
In our study, we found that the ambient aerosol size distributions measured with the SMPS
varied considerably at Table Mountain, as seen in Fig. 11, while the Plantower normalized
reported size distribution changed very little. Invariant Plantower size distributions were also
observed during controlled laboratory studies (He et al., 2020; Kuula et al., 2020; Tryner et al.,
2020). This suggests that the values in the channels above CH1 are software generated and
indicates that the most relevant output from the PMS is from the CH1 channel. The bottom row
of Table S5 shows that the PMS bin fractions above 1 µm increased by only a factor of 2–5 in a
high-PM$_{2.5}$ windblown dust episode at Keeler, California.  This is consistent with the PMS model
prediction that PMS coarse aerosol response is small relative to a perfect nephelometer.
The results above indicate that CH1 is the primary source of aerosol information from the PMS
sensor. Additionally, consistent with the sensor behaving like a cell-reciprocal nephelometer, it
was found that CH1 was proportional to the aerosol scattering coefficient, not the number
concentration of particles having diameters greater than 0.3 µm.  CH1 was approximately a
factor of 10 lower than the DMPS number concentration for a similar size range.
**5.2.7 Relationship between CH1 and PM$_{2.5}$**
The PM$_{2.5}$ mass concentration was not measured by Federal Reference Method (FRM) or Federal
Equivalent Method (FEM) instruments at MLO and BOS during this study. Consequently, the
PA-PMS PM$_{2.5}$ or CH1 results cannot be compared with PM$_{2.5}$ concentrations, but they can be
compared with measured scattering coefficients and discussed in the context of mass scattering
efficiency, which ties scattering coefficient to mass concentration.
Figure S19 shows that the PA-PMS PM$_{2.5}$ channel is reasonably well correlated with b$_{sp1}$ for
values greater than about 10–20 µg m$^{-3}$, typical of many moderately polluted locations, with a
calculated mass scattering efficiency of approximately 2.5 m$^2$ g$^{-1}$.  This value of the mass
scattering efficiency is at the low end of the range of values reported by Hand and Malm (2007),
which could reflect the nature of the observed aerosols or an error of the PA-PMS PM$_{2.5}$ mass
concentration. This suggests that the effectiveness of the PA-PMS to serve as a PM$_{2.5}$ mass
concentration monitor is due both to the sensor behaving like an imperfect integrating
nephelometer and to the mass scattering efficiency of ambient PM$_{2.5}$ aerosols being roughly
constant with values in the 2–4 m$^2$ g$^{-1}$ range.  However, it is likely that the PA-PMS
underestimates PM$_{2.5}$ for very clean areas where b$_{sp1}$ is often less than 10 Mm$^{-1}$. For example, the
PA-PMS PM$_{2.5}$ was zero for 1099 of the hours in this study when b$_{sp1}$ was greater than 1 Mm$^{-1}$.
One may obtain a lower bound estimate of the PA-PMS RMSE 1 h average mass concentration
from the study results.  Figure 14 shows the PurpleAir scattering coefficient RMSE as a function
of the measured scattering coefficient. For example, the PurpleAir NRMSE is 20% for a fine
aerosol scattering coefficient of 25 Mm$^{-1}$. For an aerosol having a mass scattering efficiency of
2–3 m$^2$ g$^{-1}$, this is approximately 10 µg m$^{-3}$.  Thus, the PurpleAir 1 h average RMSE is roughly 2
µg m$^{-3}$. This is somewhat lower than the reported mean absolute error of ~4 µg m$^{-3}$ for hourly
average PM$_{2.5}$ in Pittsburgh (Malings et al., 2020).  This error assumes that the mass scattering
efficiency is fixed and known.  This is generally not the case, and the actual error in the mass
concentrations will be larger.
The mean 1 h average fine aerosol scattering coefficient b$_{sp1}$ at MLO during our yearlong study
was 1.50 Mm$^{-1}$. From Fig. 14, the PurpleAir had a RMSE of 0.60 Mm$^{-1}$.  For an aerosol having a
mass scattering efficiency of 2–3 m$^2$ g$^{-1}$, this corresponds to a 1 h average RMSE of roughly 0.2–
0.3 µg m$^{-3}$. This is well below the advertised 1 h average MDL of commercial PM$_{2.5}$ monitors.
For example, the BAM 1020 specifies a typical hourly detection limit of 3.6 µg m$^{-3}$.

## 6. Summary, discussion, and future work

We have demonstrated that the PMS sensor inside the PA monitor (PA-PMS) appears to behave as an imperfect reciprocal integrating nephelometer. As a scattering sensor, the PMS cannot directly count nor size particles in the air stream. The PMS uses an unknown algorithm to convert the scattering signal to a near-constant normalized number distribution from which PM concentrations are derived.

The scattering coefficient that is measured by an ideal integrating nephelometer does not need correction for any aerosol attributes such as shape, chemical composition, refractive index, or diameter. It is a valuable measure for visibility and global climate monitoring. Simple low-cost sensors such as the PA-PMS can play a role in estimating aerosol scattering coefficients and improving global coverage. Yearlong field data at NOAA's Mauna Loa Observatory and Boulder Table Mountain sites show that the 1 h average of the PA-PMS CH1 is highly correlated with a nephelometer-measured fine aerosol scattering coefficient at 550 nm, $b_{sp1}$, over a wide scattering coefficient range of 0.4 $Mm^{-1}$ to 500 $Mm^{-1}$. The relationship between CH1 and $b_{sp1}$ at 550 nm is found to be $b_{sp1}$ ($Mm^{-1}$) = 0.015 × CH1 when both quantities are adjusted to the same temperature and pressure.

The physical-optical model developed in this paper for the PMS and the general consistency with both published laboratory data for a variety of fine aerosols and ambient field data may motivate users of other low-cost sensors to develop similar models. It is possible that some of the other low-cost sensors also use polarized lasers in a cell-reciprocal configuration like the PMS. Such models would improve the understanding of sensor operation and help users better recognize the opportunities and limitations of other low-cost sensors in applications such as monitoring the scattering coefficient.

The strong relationship between $b_{sp1}$ and CH1 and the general agreement between the model and published laboratory data support characterizing the PA-PMS as an imperfect truncated cell-reciprocal nephelometer. The results demonstrate that it is possible to use the PA-PMS to estimate the 1 h average fine aerosol scattering coefficient across a wide range of aerosol scattering concentrations, provided the aerosol median scattering diameter is between 0.26 μm and 0.46 μm. The CH1 and $b_{sp1}$ relationship is dependent on the size distribution, and it is expected to change for locations and times where the particle size shifts to larger or smaller sizes than those measured at BOS and MLO.

We found that the PA-PMS has important limitations compared to integrating nephelometers. It measures the light scattering over a smaller angular range, causing a significant truncation of the scattering signal in the forward and backward directions. Additionally, the PA-PMS uses a polarized light source; the sensor most likely does not have a cosine response; the laser beam profile is not a simple plane wave; and the inlet/geometry creates a broad uncertain particle size cut point. Nephelometers calibrate their scattering coefficient with $CO_2$ or Suva, but the PMS is unresponsive to these gases. As a result, there is currently no convenient way to calibrate the PMS to ensure its accuracy. Neither PA nor Plantower provide technical support. Quality assurance and control are not as robust as one encounters for regulatory and scientific monitoring instruments. For this reason, it is useful to test the PMS sensors in filtered air before using them and to limit field use to those sensors that have 1 h average CH1 values less than 2. While

sampling, it is necessary to compare 1 h averages from the two PMS sensors in each PA monitor
to become aware of any changes and, if needed, to replace them in a timely fashion.
This study limited its findings to low-RH air, because both the PA monitors and the
nephelometers were heated to reduce RH. Since RH plays such an important role in water uptake
by hygroscopic aerosols and the concomitant increase in the scattering coefficient, future work is
planned to compare unheated PA monitors with an unheated nephelometer that does not reduce
RH before sampling.  Our model predicts that the PMS may not be as responsive to hygroscopic
growth as an unheated nephelometer. This is a topic of current study.
The PA-PMS reports a mass concentration of $PM_{2.5}$ particles, and many papers have been written
to compare the PA-PMS values with reference instruments and explain the observed differences.
The modest agreement that has been reported is the direct result of two factors generally
overlooked in those publications: the PA-PMS behaves like an imperfect integrating
nephelometer that provides a representative value of the light scattering coefficient, and the mass
scattering efficiency of $PM_{2.5}$ aerosols is roughly constant, with values in the 2–4 $m^2$ $g^{-1}$ range.
**Appendix**
The PMS physical-optical model makes some simplifying assumptions. The actual PMS laser
beam profile is not a simple plane wave but complex in shape. The model assumes the laser is a
plane wave with a constant laser beam irradiance profile. This allows the use of Mie theory to
predict the light scattered by particles in the laser. Secondly, the model calculates the light
scattered to a narrow strip across the middle instead of the entire photodiode. It assumes that the
irradiance received by the narrow strip is representative of the entire photodiode.
The intensity of light scattered from a particle in the laser is
$I(\theta) = F_{dv}\, \beta(\theta)\, dv$                                          (A1)
where $I(\theta)$ is the intensity of light at angle $\theta$ scattered from a particle in the volume element dv
(with units of Watt $sr^{-1}$); $\beta(\theta)$ is the volume scattering function ($m^{-1}$ $sr^{-1}$); $F_{dv}$ is the incident laser
flux density (Watt $m^{-2}$) impinging on the volume element dv; and dv is the volume element
within the laser.
The volume scattering function for a monodisperse aerosol having a diameter $D_p$ and number
concentration $N(D_p)$ in the PMS laser is
$\beta(m, \lambda, \theta, D_p) = (\lambda/2\pi)^2\, N(D_p)\, |S_1(m, \lambda, \theta, D_p)|^2$                     (A2)
where $|S_1(m, \lambda, \theta, D_p)|^2$ is the perpendicular scattering intensity function; $\lambda$ is the laser
wavelength; m is the particle complex refractive index; $\theta$ is the scattering angle; and $D_p$ is the
aerosol diameter. Note that $\theta = 0$ in the direction of the laser, and $\theta = 90$ degrees perpendicular to
the laser and photodiode.
For one particle of size Dp in the volume element dv, $N(Dp)\, dv = (1/dv) \times (dv) = 1$.
The incremental power dP (Watt) scattered from a particle in the volume element dv across a
solid angle $d\Omega$ subtended on the surface of a sphere at distance r from the particle, and normal to
r, is
$dP = I(\theta) \, d\Omega.$ (A3)
$d\Omega = dA_0 / r^2$, where $dA_0$ is the incremental area on the sphere at distance r from the particle and
normal to r. dP is then
$dP = I(\theta) \, dA_0 / r^2.$ (A4)
For the PMS model, $dA_0$ is a small rectangle with width w and height $rd\theta$, where w is the width
of the strip on the photodiode, and $d\theta$ is the differential scattering angle.
$dA_0 = r \, d\theta \times w$, where w is the width of the strip on the photodiode. From Fig. A1, $r = b/\sin(\theta)$,
where b is the distance from the laser to the photodiode.
$d\Omega = dA_0 / r^2 = (r \, d\theta \times w)/ r^2 = d\theta \times (w/r) = (w/b) \times \sin(\theta)d\theta.$ (A5)
The incremental power across the solid angle $d\Omega$ normal to r is then
$dP = I(\theta) \times dA_0/ r^2 = I(\theta) \times (w/b) \times \sin(\theta)d\theta.$ (A6)
Substituting for $I(\theta)$,
$dP(g,x,\theta) = [F_0 \, (\lambda/2\pi)^2 \, |S_1(\theta,D_p)|^2] \times (w/b) \times \sin(\theta) \, d\theta.$ (A7)
Equation A7 can be further simplified by combining the constants into $K = (\lambda/2\pi)^2 \, F_0 w/b$, where
K has units of watts:
$dP(g,x,\theta) = K \, |S_1(\theta,D_p)|^2 \sin(\theta) \, d\theta.$ (A8)
The power received by the photodiode from a particle of diameter $D_p$ in the volume element at x
is obtained by numerically integrating across $\theta$ on the photodiode:
$P(m,D_p) = K\int_{\theta1(x)}^{\theta2(x)} \, |S_1(m,\theta,D_p)|^2 \sin (\theta) \, d\theta.$ (A9)
Due to the PMS geometry, the upper and lower angular scattering integration limits for $\theta$ depend
on the location x. This can be seen in Fig. S4. For example, at $x = 0$ mm, the upper and lower
integration limits for $\theta$ are 18 to 38 degrees. At $x = 4.0$ mm, over the center of the photodiode,
the angular integration limits are 50 to 130 degrees.
The total power P in Watts received by the photodiode from the light scattered by all the
particles of diameter $D_p$ in the laser is obtained by carrying out the numerical integration in Eq.
A9 for all x from 0 to 10 mm:
$P(m,D_p) = K \int_{x=0}^{x=10mm} \int_{\theta 1(x)}^{\theta 2(x)} |S_1(m,\theta,D_p)|^2 \sin(\theta) \, d\theta \, dx.$          (A10)
The result for carrying out this calculation for the power per particle of size $D_p$ is in Table S6 for
wavelength 657 nm and particle refractive index 1.53+0.015i. The total power received at the
photodiode by a distribution of particles is obtained by summing up the power per particle of size
$D_p$ times the number of particles $N(D_p, m)$ in the size interval $D_p$ to $D_p + dD_p$.
$P = K \int_{Dp} \int_{x=0}^{x=10mm} \int_{\theta 1(x)}^{\theta 2(x)} |S_1(m,\theta,D_p)|^2 \sin(\theta) \, N(D_p, m) \, d\theta \, dx \, dD_p.$          (A11)
Figure A1 shows the PMS geometry. The distance along the laser is the variable x, which ranges
from 0 to 10 mm. The distance along the photodiode is the variable g, which ranges from 0 to 3.0
mm. The distance between the photodiode and the laser is b, approximately 1.8 mm.

## Code and data availability

Data sets are available on Zenodo.org.  The dataset is available at
https://doi.org/10.5281/zenodo.5764982.  The data were published on 15 December 2021.
Citation is: Ouimette, James; Malm, William; Schichtel, Bret; Sheridan Patrick; Andrews,
Elisabeth; Ogren, John A.; & Arnott, W. Patrick. (2021). Datasets for paper "Evaluating the
PurpleAir monitor as an aerosol light scattering instrument" (1.0_20211215) [Data set]. Zenodo.
https://doi.org/10.5281/zenodo.5764982

## Supplement

I think ACP fills out this - we have provided the supplement to them.

## Author contributions

PJS and JRO designed the field study. PJS provided measurement data from the MLO and BOS
sites. BAS performed the uncertainty and precision analyses. WCM led the physical-optical
model development with BAS and JRO. JRO and WCM performed analysis of laboratory and
field data for model evaluation. WPA provided insight regarding the laser behavior and
instrument electronics. JAO and WCM independently performed the statistical analyses of the
TSI nephelometer and PA data. JAO and JRO performed measurements and tests to characterize
the PMS5003. EA led the manuscript submission and review process. All authors read and
commented on the article.

## Competing interests

The authors declare that they have no conflict of interest.

## Acknowledgements

The authors acknowledge the following for their contributions:
Jim Wendell, NOAA, for engineering support, Marty Martinsen, NOAA, for MLO field support,
Derek Hageman, CIRES, University of Colorado, for data acquisition and processing support,
and Helene Bennett, CIRA, Colorado State University, for technical editing support. The

National Park Service participation in this project was supported under the cooperative agreement P17AC00971.

## Disclaimer

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

**Tables**

**Table 1.** Summary of PA, TSI nephelometer, and DMPS data coverage.

| | number of hours | | | | percent coverage | | | |
| --- | --- | --- | --- | --- | --- | --- | --- | --- |
| site | PA-PMS | TSI neph | DMPS | overlap | PA-PMS | TSI neph | DMPS | Time period |
| **MLO** | 9371 | 9204 | na | 9204 | 97.6 | 95.9 | na | 2019-05-06 to 2020-06-05 |
| **BOS** | 7716 | 7479 | 7045 | 6901 | 97.7 | 94.7 | 89.2 | 2020-02-13 to 2021-01-06 |

**Table 2.** Summary of PA-PMS and nephelometer hourly observations at MLO and BOS.

| | 1 h median (average) | | | | 1 h range min-max | | | |
| --- | --- | --- | --- | --- | --- | --- | --- | --- |
| | PA-PMS | | TSI nephelometer | | PA-PMS | | TSI nephelometer | |
| **Site** | $PM_{2.5}$ µg m$^{-3}$ | CH1 | $b_{sp10}$ Mm$^{-1}$ | $b_{sp1}$ Mm$^{-1}$ | $PM_{2.5}$ µg m$^{-3}$ | CH1 | $b_{sp10}$ Mm$^{-1}$ | $b_{sp1}$ Mm$^{-1}$ |
| **MLO** | 0.000 (0.12) | 26.7 (75.2) | 1.19 (2.82) | 0.76 (1.50) | 0.0 - 21.6 | 0.26 - 1649 | -0.35 - 35.2 | -0.29 - 34.2 |
| **BOS** | 3.37 (8.42) | 720 (1422) | 14.6 (32.4) | 9.9 (20.9) | 0.0 - 571 | 7.38 - 63340 | -0.11 - 4097 | -0.44 - 2596 |

**Table 3.** Ordinary least square regression coefficients with a zero intercept and standard error for $b_{sp1}$ and CH1 as the dependent and independent variables, respectively, for the BOS, MLO, and combined datasets. CH1 and $b_{sp1}$ reported at STP. Also shown are the respective coefficients of determination, $R^2$.

| Site | slope (Mm$^{-1}$) | standard error (Mm$^{-1}$) | 1/slope (Mm) | $R^2$ |
| --- | --- | --- | --- | --- |
| **BOS** | 0.015 | $2.68 \times 10^{-5}$ | 67.0 | 0.97 |
| **MLO** | 0.017 | $5.72 \times 10^{-5}$ | 59.0 | 0.85 |
| **Both BOS&MLO** | 0.015 | $2.07 \times 10^{-5}$ | 67.0 | 0.97 |

**Figures**

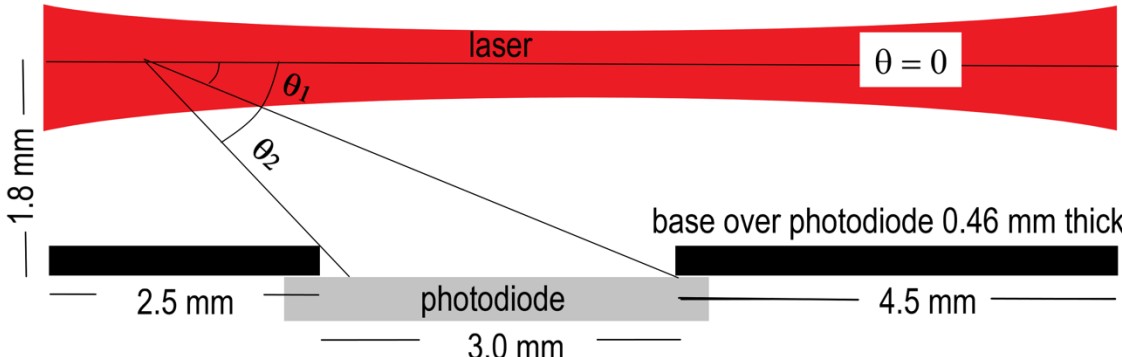


**Figure 1.** PMS sensor geometry highlighting the dimensions of laser beam (red) and photodiode
(gray) and the various relevant distances between the two.


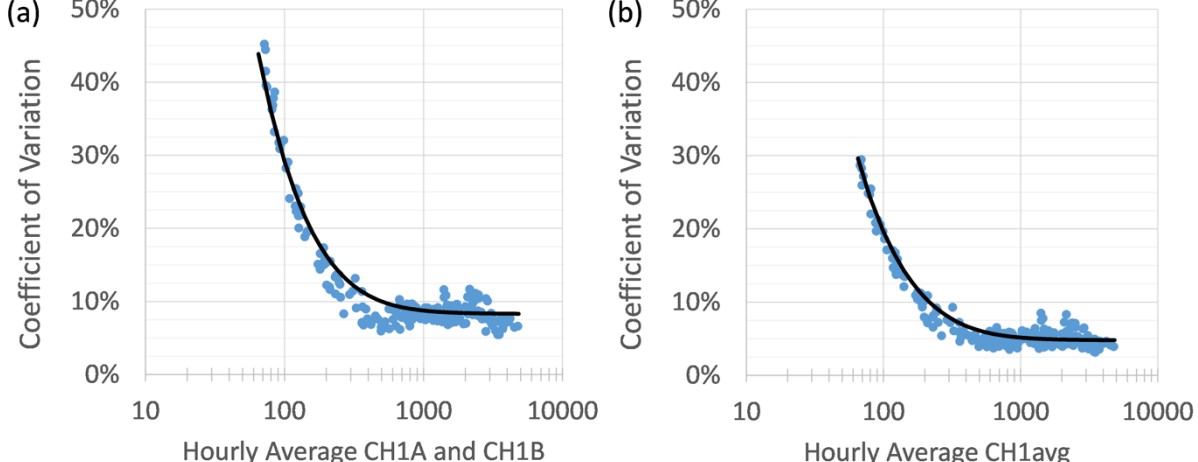


**Figure 2.**  Precision estimated as the coefficient of variation of the hourly CH1A and CH1B (a)
and CH1avg values (b) for the 19 collocated sensors and 9 PAs.

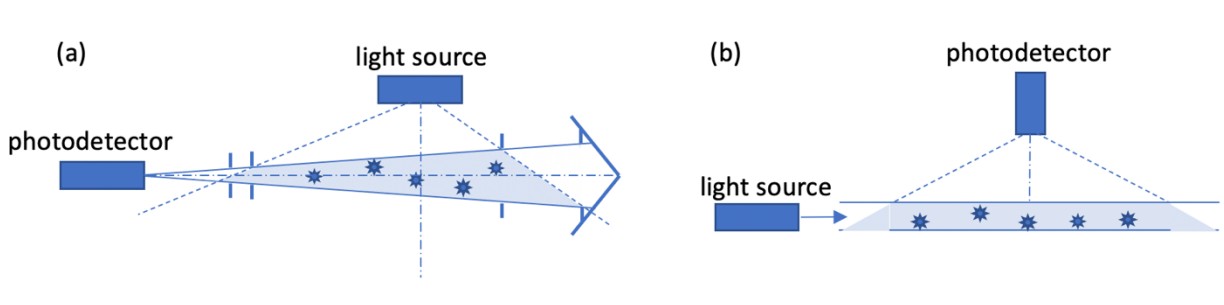


**Figure 3.** Diagrams of the (a) cell-direct nephelometer and (b) cell-reciprocal nephelometer.


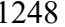

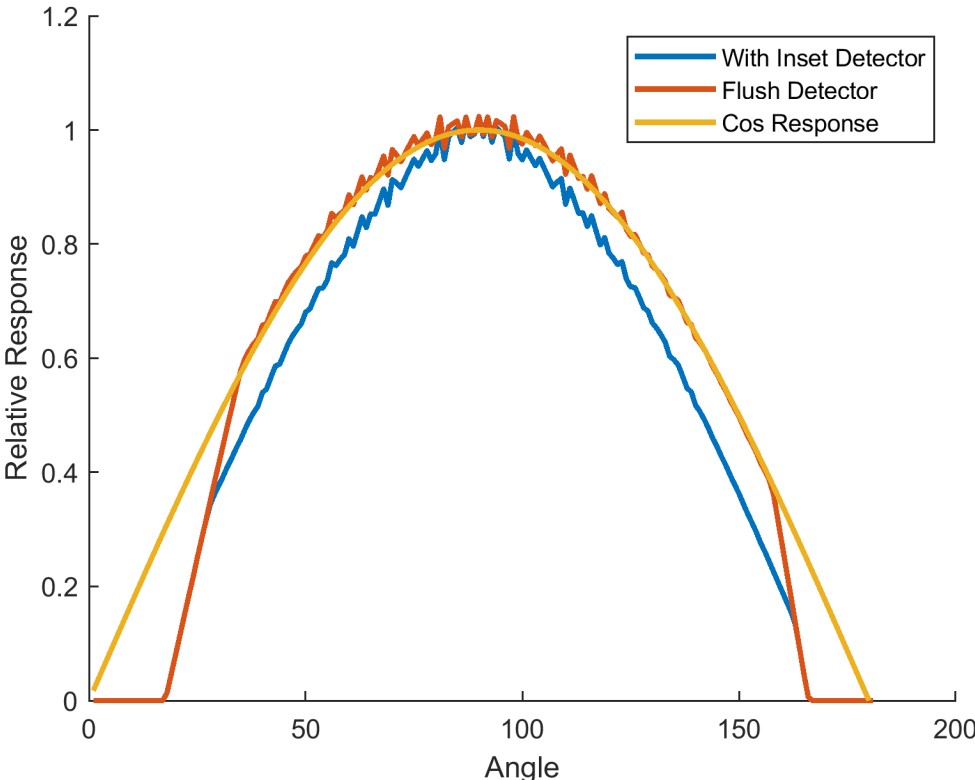


**Figure 4.** Relative response of the photodetector resulting from truncated scattering angles and a recessed photodetector. See explanation of the different curves in text.


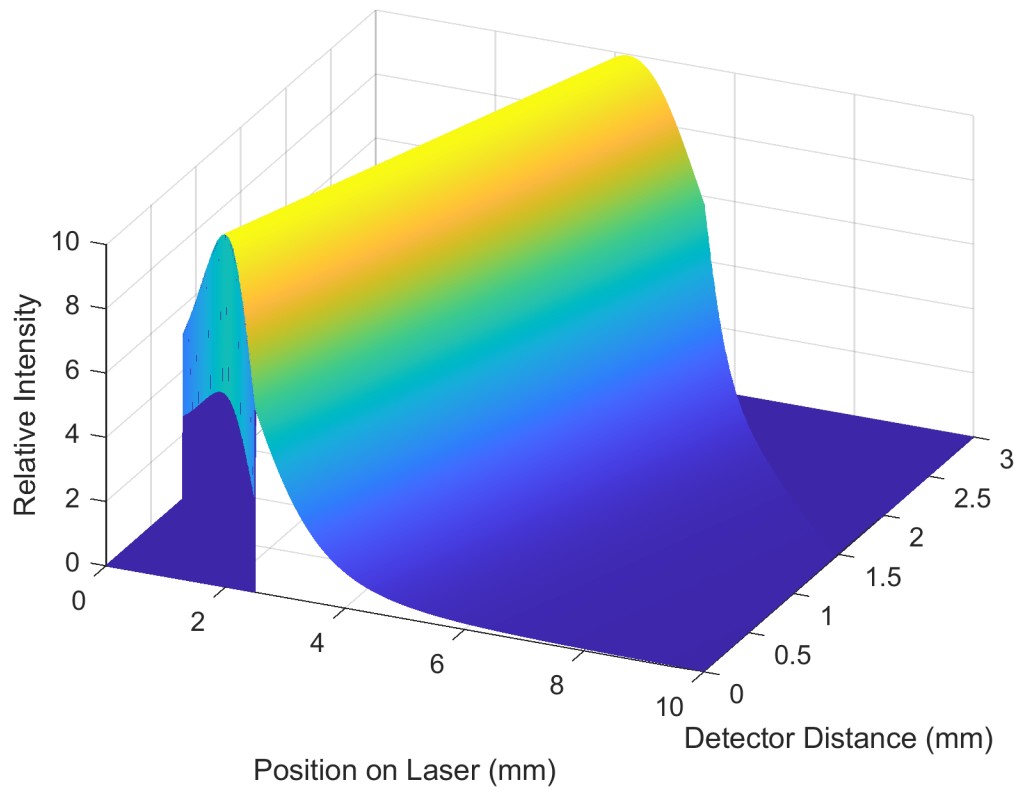


**Figure 5.** Relative intensity of radiant energy scattered by a lognormally distributed aerosol
volume size distribution with a volume mean diameter of 0.33 μm and geometric standard
deviation of 1.7 as a function of location of scattering event in the laser and as a function of
position on the photodiode.  Assumed laser wavelength was 650 nm, and the particle index of
refraction was assumed to be 1.53.  Positions in the laser and detector are from left to right as in
Fig. 1 and are in units of mm.

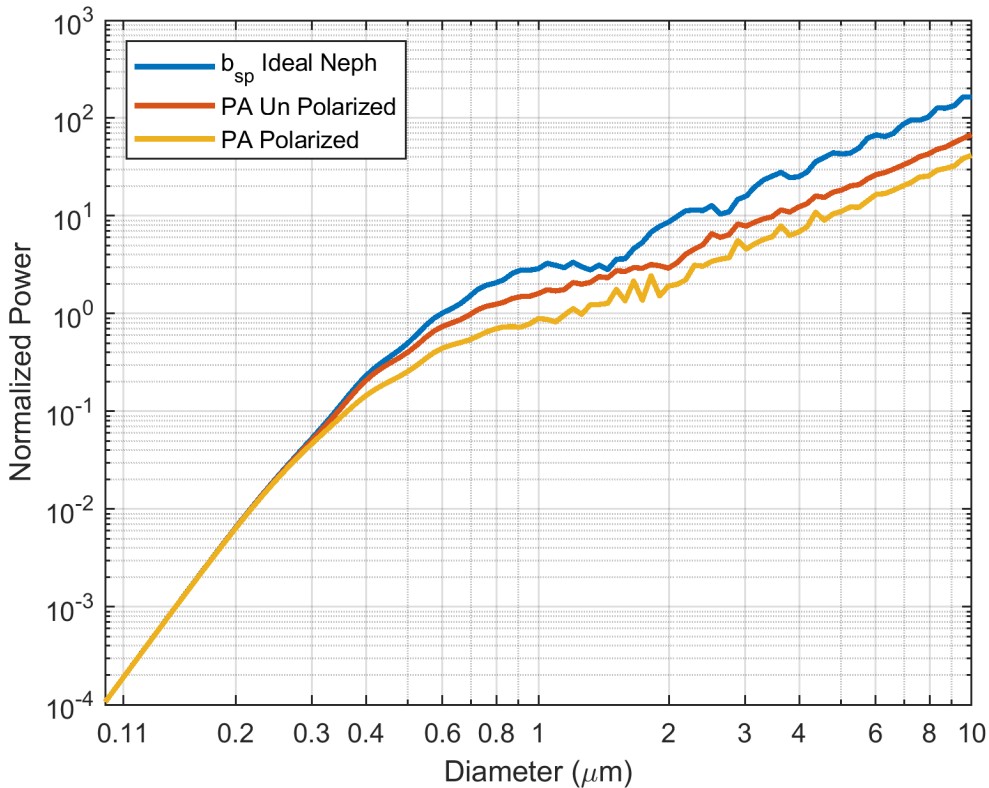


**Figure 6.** Normalized power detected by an ideal integrating nephelometer, a PMS with an
unpolarized light source, and a PMS with a perpendicularly polarized light source plotted as a
function of particle diameter. Modeled light source wavelength is 657 nm, and the particle index
of refraction is 1.53. See explanation of the different curves in text.

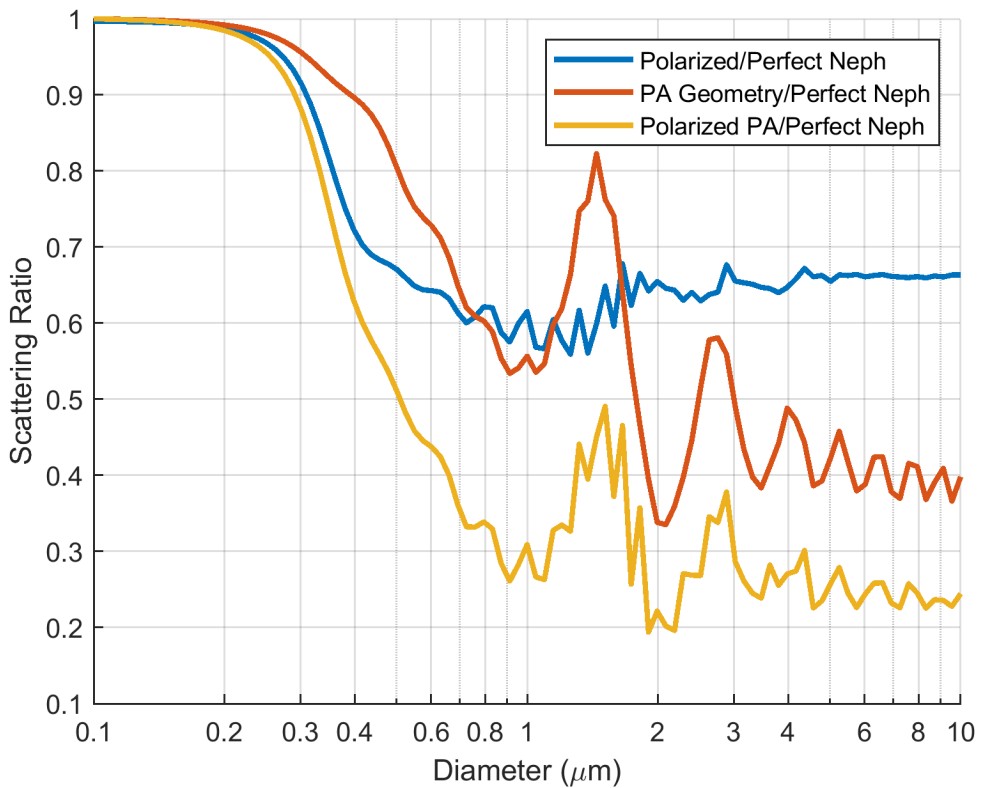


**Figure 7.** Ratio of scattering of a "perfect" nephelometer to a nephelometer with a light source
that is perpendicularly polarized (blue) and to a PMS with an unpolarized light source (red).
Yellow line shows the effect of a perpendicularly polarized light source and PMS geometry. All
three curves are plotted as a function of particle diameter.


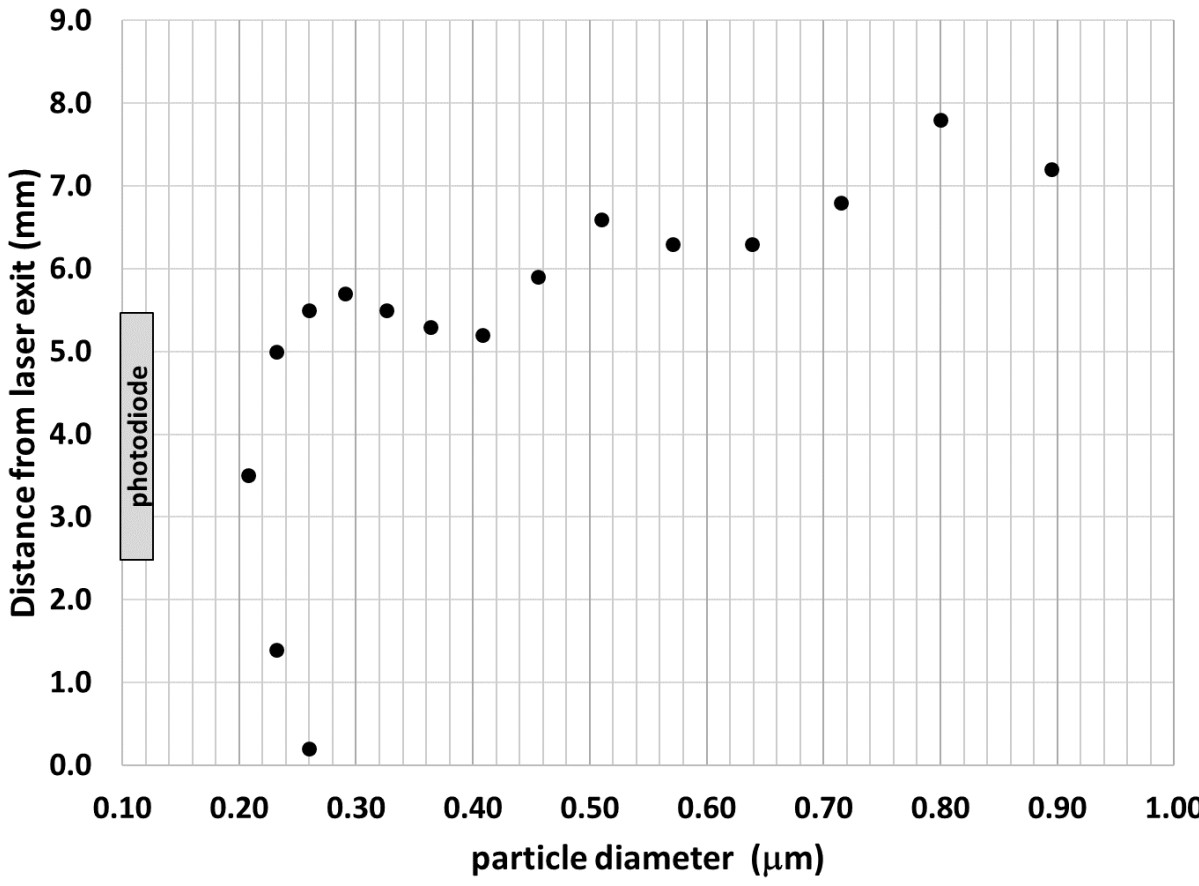


**Figure 8.** The model predicts that different size particles can generate the same irradiance on the photodiode, depending on their location in the laser beam. In this example, each of the particles would create $1.7 \times 10^{-2}$ picowatts of scattered irradiance on the photodiode.



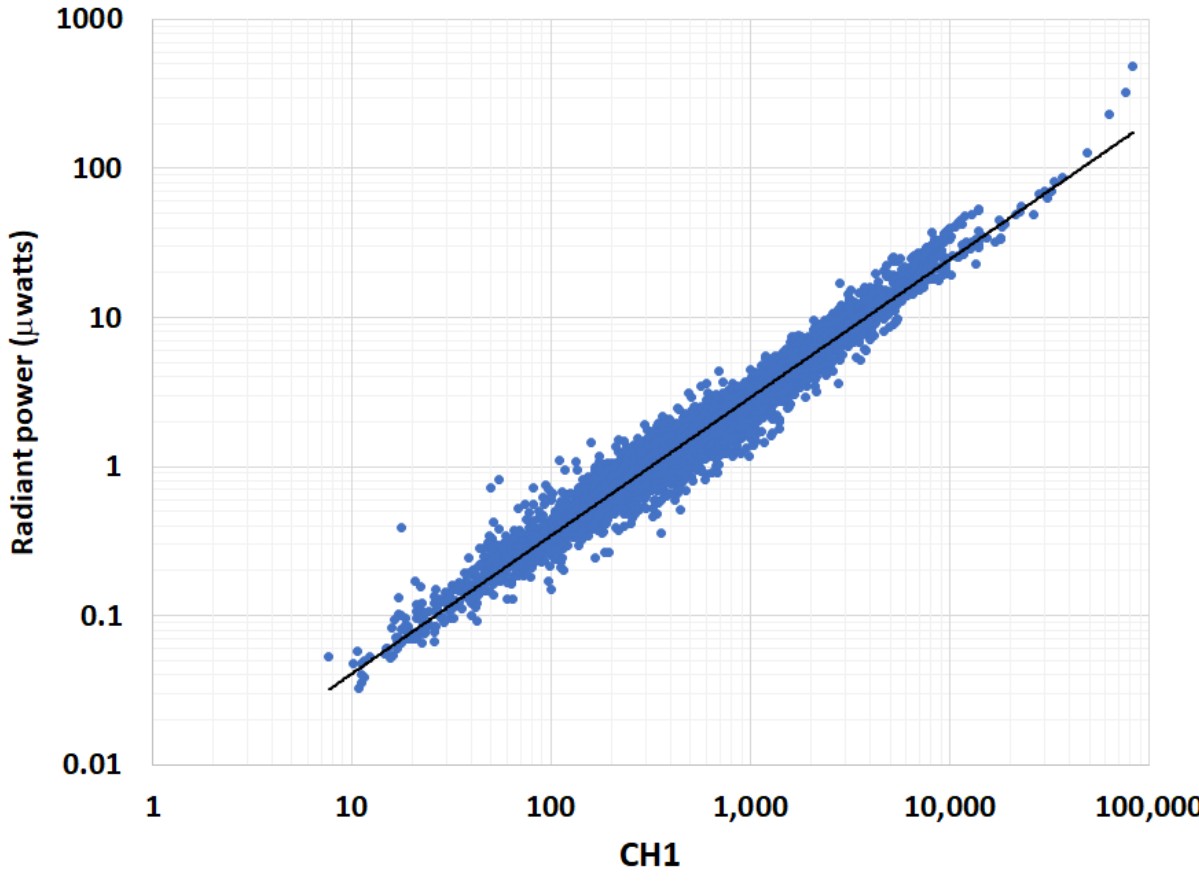


**Figure 9.** One hour average CH1 reported values plotted against model-predicted radiant power
(or energy) in µwatts on the photodiode. Both the CH1 and DMPS data were adjusted to STP
conditions. Ordinary least squares regression line is also shown.  Plot is based on 6839 1 h
averages at BOS.

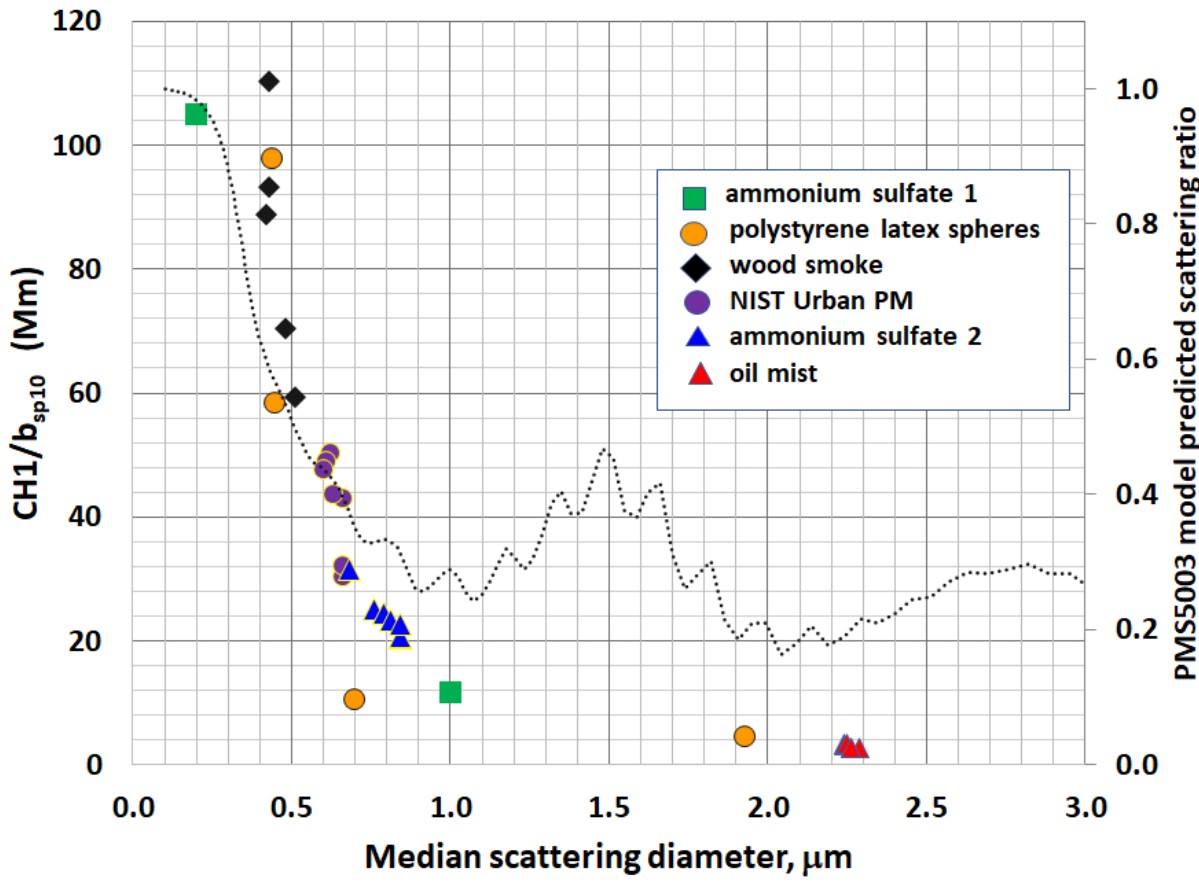


**Figure 10.** Laboratory measurements of CH1/$b_{sp10}$ versus median scattering diameter from
Tryner et al. (2020) and He et al. (2020).  Results are compared with PMS physical-optical
model prediction of the scattering ratio (yellow line in Fig. 7). Maximum value of 1.0 of the
model-predicted scattering ratio is arbitrarily set at a CH1/$b_{sp10}$ value of 110 Mm. Ammonium
sulfate 1 data are from He et al. (2020); all other data are from Tryner et al. (2020).

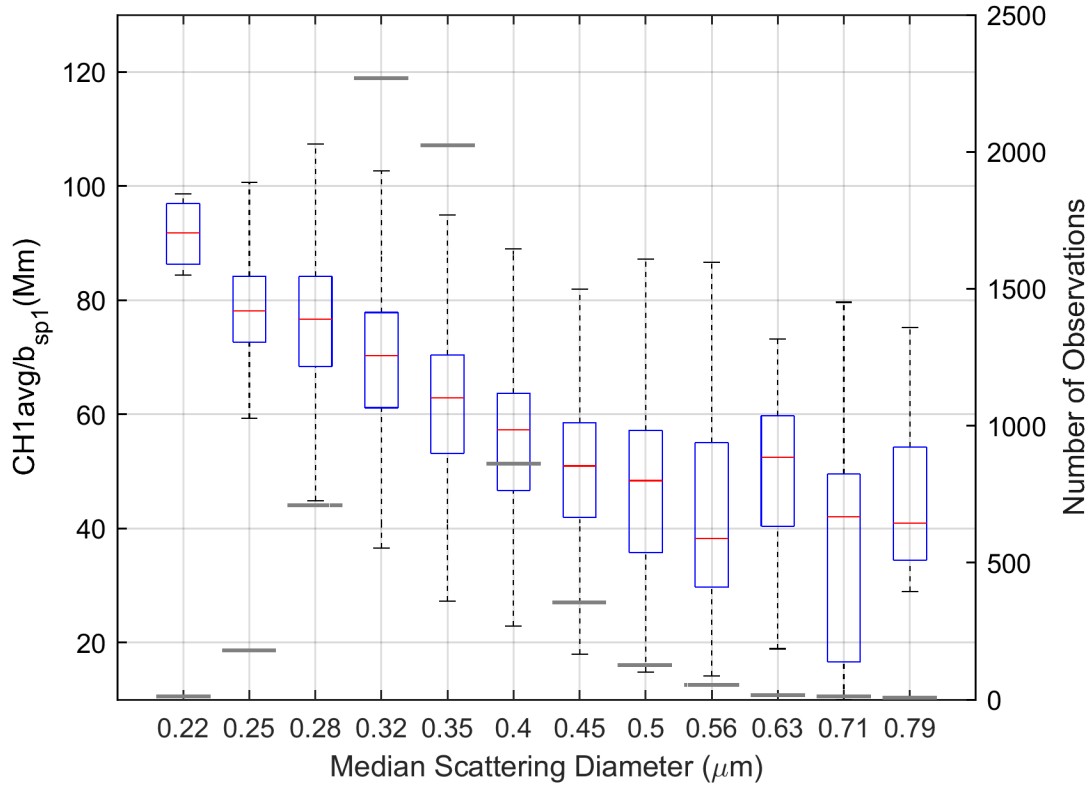


**Figure 11.** Observed decrease in CH1avg/$b_{sp1}$ ratio as a function of MSD values. MSD values
were selected based on a log scale but plotted equally spaced from each other to maintain
uniformity in the dimensions of the box and whisker symbol. Red line represents the median
value, and the bottom and top of each box are the first and third quartile values. Extremes shown
on each box are the 2 and 98 percentiles. Black horizontal lines for each MSD value are the
number of observations in the respective MSD bins.


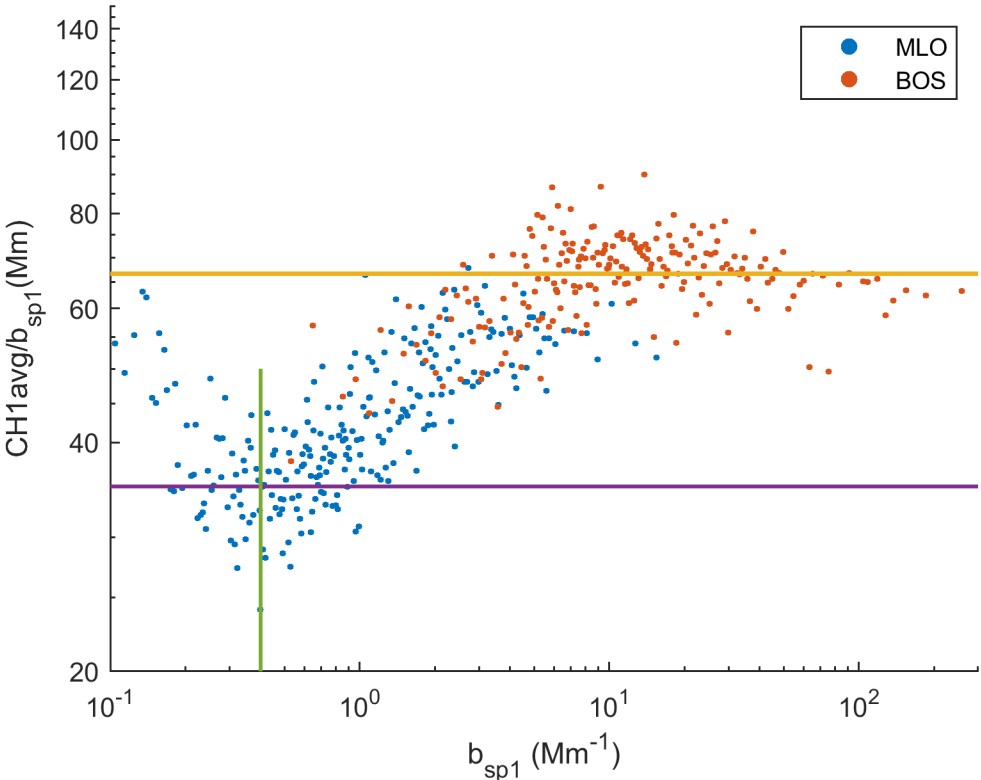


**Figure 12.** Ratios of CH1avg and measured scattering, $b_{sp1}$, as a function of measured $b_{sp1}$ for
MLO and BOS. Green line corresponds to 0.4 $Mm^{-1}$ while the purple line, a ratio of 35 Mm,
corresponds to the additive uncertainty of 14. Yellow line corresponds to a CH1avg/$b_{sp1}$ ratio of
approximately 67 Mm, the slope of CH1avg vs. $b_{sp1}$ above about 5 $Mm^{-1}$.



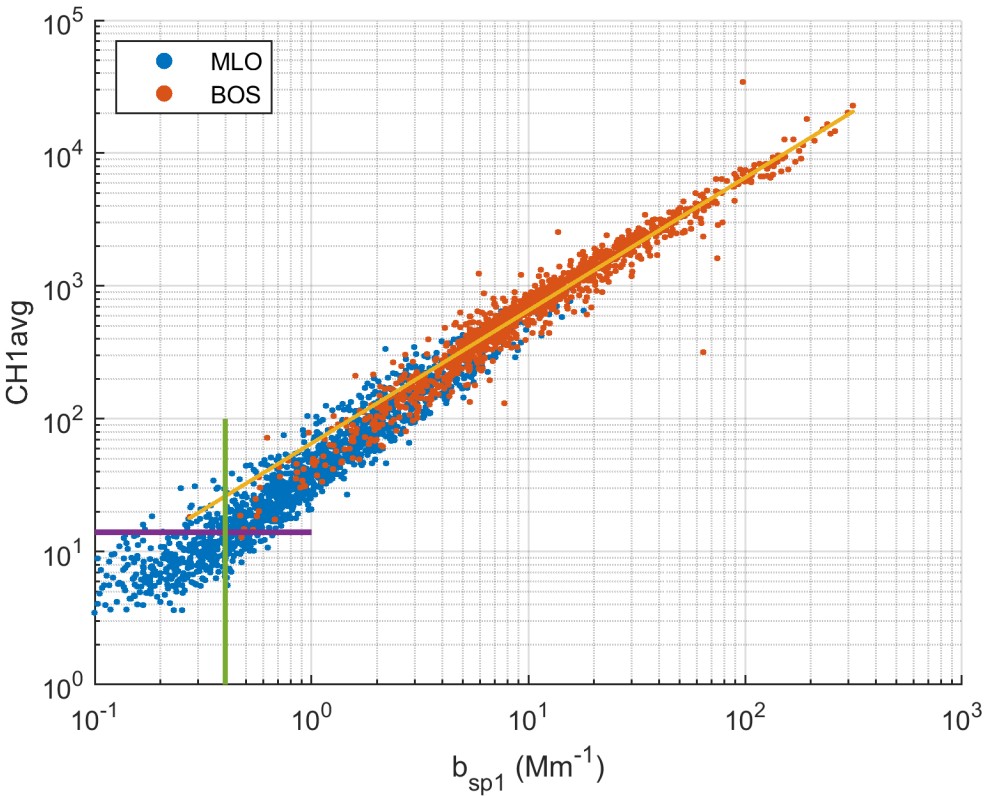


**Figure 13.** Fine aerosol scattering coefficient from TSI nephelometer vs. CH1avg value from
PA. Yellow line represents the fit to all data. Purple line shows the additive uncertainty of 14
while $b_{sp1}$ values less than the green line were removed for the regression analysis.

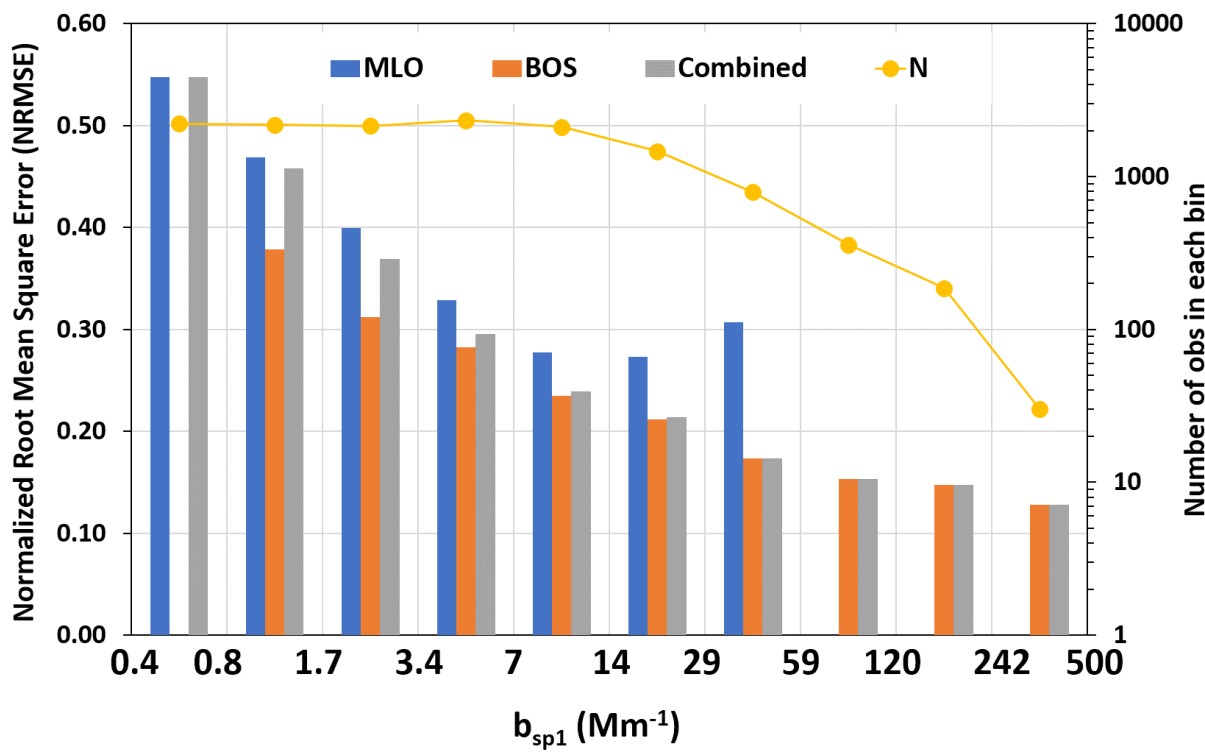


**Figure 14.** Normalized root mean square error between measured and estimated scattering from
CH1 values plotted as a function of binned $b_{sp1}$ for the BOS, MLO, and combined datasets.
Yellow line is referenced to the right axis to provide the number of observations in each bin.
Numbers on the x-axis represent the lower and upper levels of each scattering bin.

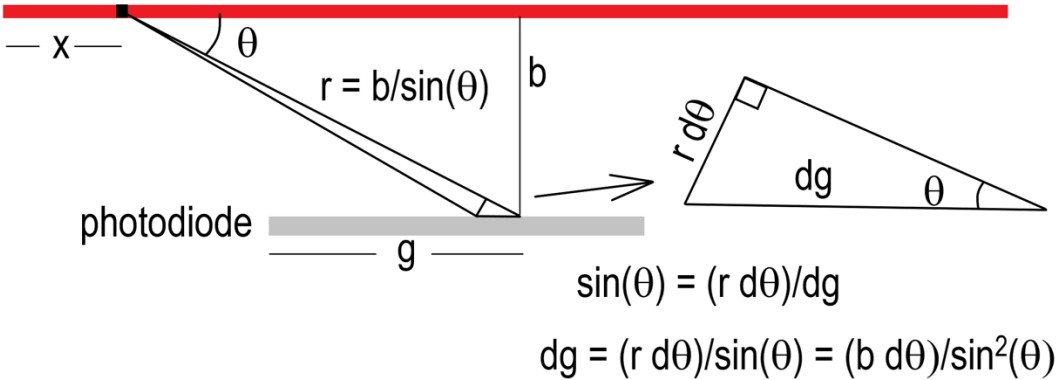


**Figure A1.** Sketch of PMS5003 geometry.