# Peer review of "Evaluating the PurpleAir monitor as an aerosol light scattering instrument"

_Atmospheric Measurement Techniques, 2021_

## Author Comment (AC1)

We would like to thank the reviewers for their clear and positive reviews. We have addressed comments below. Our responses are in blue. Changes to the manuscript text where needed are noted in red.

Some things to alert reviewer to:

Abstract is slightly rearranged and has some new text reflecting changes in the manuscript.

We have made some very minor updates to the text to add a clarifying word or sentence in places. These changes are noted in red in the text. We have also made some minor adjustments to some of the figures (e.g., changed um to μm on axes labels).

Added 2 new sections:

- Section 5.2.2 "Predicted aerosol size truncation versus published laboratory data" applies the approach shown in original Sect. 5.2.2 (which is now Sect. 5.2.3) for BOS field observations to laboratory results reported in the literature. This section uses much of the text of the original Sect. 5.2.2 but adds discussion related to comparison with published laboratory results. This section contains a new figure - Fig. 10 in revised manuscript - comparing laboratory observations from literature to our model. Also addresses comment #4.
- Section 5.2.7 "Relationship between CH1 and $PM_{2.5}$" breaks out the brief discussion of this relationship in the original Sect. 5.2.5 and slightly expands it to tie it to mass scattering efficiency information. Also addresses comment #8.

Added cited references to end of supplement.

**Reviewer#1 - R. Subramanian**

The PurpleAir (and similar low-cost PM) sensors have received a lot of attention recently, due to their popularity and wide deployments worldwide. Many publications have shown the utility of these sensors for air quality applications, including (at the risk of tooting my own horn) several of my own papers (e.g. Rose Eilenberg et al. https://doi.org/10.1038/s41370-020-0255-x). A number of laboratory and field evaluations have shown the limitations of these sensors, especially the fact that the size distribution reported by the Plantower sensor is unrealistic (e.g. Kuula et al., Tryner et al., Zou et al.) and that the Plantower sensor is a nephelometer, not a particle counter despite manufacturer claims (He, Kuerbanjiang, & Dhaniyala). Bernd Laquai (2017) had shown the lack of response of a similar low-cost PM sensor to coarse aerosols https://www.researchgate.net/publication/320707986_Particle_mass_distribution_dependent_inaccuracy_of-low_cost_sensors_Bjhike_HK-A5 ; I have seen similar results in our testing in Doha, Qatar (hopefully, to be published soon!)

Hagan and Kroll have also published an optical model for various low-cost OPCs https://amt.copernicus.org/articles/13/6343/2020/amt-13-6343-2020.html

However, this paper makes a unique contribution in that they measure the insides of the sensor, develop a nice model specific to the PurpleAir PMS5003 that predicts the nephelometric response compared to a perfect nephelometer, and use that to show that the

Plantower effectively cannot detect supermicron particles due to the optical configuration, with a smaller contribution from the flow path (which was my default assumption). These results likely also explain the laboratory test results across a range of PM sensors (Plantower, Nova, Sensirion, etc.) by Kuula et al. (though the Omron remains a mystery - it uses a thermal heater for convective flow with a relatively straight path, and was responsive to coarse mode aerosol). The long-term evaluation at Mauna Loa and Boulder further enhance the value of this paper. Hence, I recommend publication after minor revisions.

As noted above, we appreciate these positive comments. I should note we have added the following sentences to the abstract and conclusions to more clearly explain why the PurpleAir has been shown to be generally correlated with $PM_{2.5}$ reference instrument observations:

In abstract: The effectiveness of the PA-PMS to serve as a $PM_{2.5}$ mass concentration monitor is due both to the sensor behaving like an imperfect integrating nephelometer and to the mass scattering efficiency of ambient $PM_{2.5}$ aerosols being roughly constant.

In Sect. 5.2.7: This value of the mass scattering efficiency is at the low end of the range of values reported by Hand and Malm (2007), which could reflect the nature of the observed aerosols or could reflect an error of the PA-PMS $PM_{2.5}$ mass concentration. This suggests that the effectiveness of the PA-PMS to serve as a $PM_{2.5}$ mass concentration monitor is due both to the sensor behaving like an imperfect integrating nephelometer and to the mass scattering efficiency of ambient $PM_{2.5}$ aerosols being roughly constant with values in the 2–4 $m^2$ $g^{-1}$ range.

In conclusions: The PA-PMS reports a mass concentration of $PM_{2.5}$ particles, and many papers have been written to compare the PA-PMS values with reference instruments and explain the observed differences. The modest agreement that has been reported is the direct result of two factors generally overlooked in those publications: the PA-PMS behaves like an imperfect integrating nephelometer that provides a representative value of the light scattering coefficient, and the mass scattering efficiency of $PM_{2.5}$ aerosols is roughly constant with values in the 2–4 $m^2$ $g^{-1}$ range.

In references: Hand, J. L., and Malm, W. C.: Review of aerosol mass scattering efficiencies from ground-based measurements since 1990, J. Geophys. Res., 112, D16203, doi:10.1029/2007JD008484, 2007.

**Specific comments:**

1. Please use $PM_{2.5}$ throughout, not the eyesore that is "PM2.5".

One person's eyesore is another person's speed typing! We've made the change (and also changed PM1 and PM10 to match).

2. Line 178: Is the 9.4 $cm_3$ volume for one half of the sensor or both halves?

This was more explicitly described in the supplemental materials Sect. S1. The 9.4 $cm^3$ is the volume on the side with the printed circuit board. The volume on the side with the laser is 1.1 $cm^3$. We have clarified this now in the main text by adding the volume of the chamber on the side with the laser and the total PMS volume:

Particles then make a 180 degree turn through three exit holes at the top of the chamber to emerge on the other side of the circuit board and flow downhill through a 1.1 cm$^3$ channel that is illuminated by the laser. The total PMS volume is estimated to be $9.4 + 1.1 = 10.5$ cm$^3$.

3. Lines 201-204: This seems a noteworthy finding - that even at low wind speeds, particles larger than 2 μm may not even make it inside the sensor - with losses of ~90% for 5 μm particles. These losses, combined with the inefficient scattering due to the optical configuration, likely explain Kuula's results. On the other hand, the Omron sensor in Kuula's testing detected coarse particles, and that flow path is more straightforward (uses convective heating for aspiration).

Yes, even at low wind speeds, particles larger than 2 μm may not make it inside the laser. Further work needs to be done to better characterize the aspiration efficiency for the Plantower PMS5003. Good point about the Omron. It's worth a follow up to explore aspiration and other differences between the Omron and Plantower sensors for the supermicron aerosol. We have noted in the abstract that aspiration plays a role in the particle losses:

This is a result of using a laser that is polarized, the angular truncation of the scattered light, and particle losses (e.g., due to aspiration) before reaching the laser.

and have included the following paragraph at the end of Sect. 2.2.1:

In summary, it is likely that the laser in the PMS is sampling a lower concentration of particles >2 μm diameter than in the ambient air. Based on the literature and calculations, the dominant coarse aerosol loss mechanism may be aspiration, not internal losses. However, further measurements are needed to assess the various aerosol loss mechanisms.

4. Line 223 - it would appear (given the aspiration efficiency above) that this statement "most particles larger than 10 μm are lost" could be made for particles larger than 2 μm?

Yes, the various semi-empirical equations in the literature suggest that the 2-μm aspiration efficiency may be substantially lower than 100% for the Plantower sensor. Please see text added to Sect. 2.2.1 described in response to previous comment.

In addition, we have added a new figure later in the paper; new Fig. 10 in new Sect. 5.2.2 shows that measured CH1/b$_{sp10}$ ratios for 2 μm PSL and 2.2 μm oil mist aerosols are significantly lower than our physical-optical model predicts, which would be consistent with 2 μm aerosol losses before laser detection. Below is the new Fig. 10 along with the complete text from new Sect. 5.2.2. As noted above, the new Sect. 5.2.2 uses much of the text of the original Sect. 5.2.2. In the text below, black text represents verbiage from the originally submitted manuscript while new text is in red:

**5.2.2 Predicted aerosol size truncation versus published laboratory data**

The PMS physical-optical model described in Sect. 3 predicts that if CH1 is proportional to the photodiode power, then its signal will be reduced relative to a perfect nephelometer. Thus, the ratio CH1/b$_{sp}$ should decrease as median scattering diameter increases. To test this prediction, data were obtained from published laboratory studies evaluating the PMS against aerosols of varying composition and size distribution reported by Tryner et al. (2020) and He et al. (2020).

These reported aerosol size distributions were used here to calculate the aerosol scattering coefficient distributions from 0.1 to 10 μm for the various aerosols and refractive indices at a wavelength of 657 nm. The median scattering diameter (MSD) was calculated for each test. The MSD is the aerosol diameter at which approximately half of the light scattering coefficient is due to particles smaller than the MSD and the other half to particles larger than the MSD. The MSD was then compared to the ratio of the measured CH1 and $b_{sp10}$ values, i.e., CH1avg/$b_{sp10}$, for each of the tests reported in Tryner et al. (2020) and He et al. (2020). Fig. 10 summarizes the results for CH1/$b_{sp10}$ versus MSD.

[Figure]

**Figure 10.** Laboratory measurements of CH1/$b_{sp10}$ versus median scattering diameter from Tryner et al. (2020) and He et al. (2020). Results are compared with PMS physical-optical model prediction of the scattering ratio (yellow line in Fig. 7). The maximum value of 1.0 of the model-predicted scattering ratio is arbitrarily set at a CH1/$b_{sp10}$ value of 110 Mm. Ammonium sulfate 1 data are from He et al. (2020); all other data are from Tryner et al. (2020).

The controlled laboratory results are in general agreement with the PMS physical-optical model, showing substantial submicron aerosol truncation for the aerosols tested. The laboratory results show even more truncation than the model predicts at diameters larger than 1 μm. This suggests the possibility of supermicron aerosol loss before laser detection, perhaps due to aspiration as discussed in Sect. 2.2.1.

Because Sect. 5.2.3 in the revised manuscript was split off from the original 5.2.2 section, we have added a new introductory sentence to improve the transition:

Although ambient aerosols may vary considerably in composition and morphology and cannot be as simply characterized as laboratory aerosols, it is instructive to evaluate if PMS angular truncation can be observed using field data.

5. Sec 3.3 - it is not clear how (or if) the model accounts for the non-ideal response of the photodiode - SI suggests it does not. Please explain how this may affect your results.

Good point. The actual photodiode model in the Plantower PMS5003 is unknown. The photodiode appearance is similar to the BPW34 silicon PIN photodiode.  In this paper the specifications of the BPW34 are used to estimate the likely properties of the detector in the PMS. We have added the following text to Sect. 3.3:

As noted in Sect. 2.2.3, the specifications of the BPW34 are used to estimate the likely properties of the detector in the PMS.  Our model assumes two ideal properties of the photodiode. The first is area uniformity - that a photon impinging any part of the photodiode would generate the same current as the same photon impinging on another part of the photodiode. The second ideal assumption is that the dependence of the photodiode current on the light intensity is very linear over 4 or more orders of magnitude.  If these assumptions do not hold, then the yellow curve in Fig. 7 will change.

6. Fig 7 - this figure might be better shown (or additionally shown) without the normalization to 0.1 μm particle. It will be interesting to know the absolute effect of polarization and PA geometry on scattering at a given diameter without this normalization.

It is not clear exactly how to interpret "absolute effect".

There are at least three scenarios that come to mind: 1) you have an unpolarized laser and then turn off the parallel polarization; 2) you have a polarized and unpolarized laser with exactly the same total power output from each independent of polarization; and 3) the instruments with and without polarization are calibrated to clean air (Rayleigh conditions). When each instrument is calibrated to clean air below about 0.1 μm, the response to aerosol diameter is linear (Rayleigh regime) so that, once calibrated, an instrument with a polarized light source will "report" the same value as one with an unpolarized light source as a function of particle diameter up to about 0.1 μm.  Figure 6 essentially shows the absolute response of a nephelometer and a PA with and without a polarized light source after being "calibrated" to a "Rayleigh" gas. The calibrated response between the instruments would be the same all the way down to 0.4 nm (the approximate size of a nitrogen molecule). Because of the ambiguity of how to interpret absolute instrumental response to a polarized or unpolarized light source, we have not added any additional graphics. We have added a line to Fig. 7 that shows the modeled ratio of PA geometry to that of a perfect nephelometer. Now Fig. 7 shows ratios for three situations: PA geometry, polarized light source, and PA geometry + polarized light source.  We feel that Figs. 6 and 7 reflect relative instrumental response under real-world situations where the nephelometers or photometers are "calibrated" to Rayleigh conditions.  Below we've pasted the new Fig. 7 and its updated caption. We've also updated the text describing the figure as follows:

To highlight the effect of polarization, the blue line shows the ratio of an ideal nephelometer with a laser light source that is perpendicularly polarized to an ideal nephelometer with an unpolarized light source while the red line shows just the effect of PMS geometry relative to an ideal nephelometer. The yellow line shows the effects that polarization and PMS geometry have on the measured scattering signal.

[Figure]

**Figure 7.** Ratio of scattering of a "perfect" nephelometer to a nephelometer with a light source that is perpendicularly polarized (blue) and to a PMS with an unpolarized light source (red). Yellow line shows the effect of a perpendicularly polarized light source and PMS geometry. All three curves are plotted as a function of particle diameter.

We have added the following text to the manuscript in Sect. 2.2.2 just to highlight the importance of recognizing polarization in modeling a sensor:

It is probable that many low-cost PM sensors have lasers that are polarized. Polarization will affect how the sensors respond to various size particles and needs to be considered when modeling sensor behavior.

We have also included the following sentence in the conclusions section:

Additionally, the PA-PMS uses a polarized light source; the sensor most likely does not have a cosine response; the laser beam profile is not a simple plane wave; and the inlet/geometry creates a broad uncertain particle size cut point.

7. Sec 4.3 - PurpleAir monitors come in a plastic housing and are not heated (except for some internal heating by the electronics board). Since Mauna Loa and Boulder are both low-RH environments usually, the effect of the heater modification introduced by the authors

is unclear. Can you provide a comparison of the RH in the heated and unheated PurpleAir with ambient RH? Does heating affect the plastic housing of the PurpleAir e.g. offgassing/melting?

We found that the Mauna Loa heater increased the temperature of the Bosch sensor inside the heated PurpleAir unit by 5–8 degrees Celsius compared to the unheated PurpleAir. We have added the following text to the manuscript:

The gentle warming of the heated PurpleAir only a few degrees above ambient, is unlikely to cause the PVC to off-gas or melt.  Heating from direct sunlight may have had a larger impact.

8. Can you compare the NRMSE values with uncertanties reported in previous PurpleAir evaluations? e.g. in Malings et al. (2020), we report a mean absolute error of ~4 µg/m3 for hourly average PM2.5 (average PM2.5 in Pittsburgh is ~10 µg/m3). Others have published similar values as well.

We can't directly compare the uncertainties in our calculated $B_{sp}$ values with published uncertainties in $PM_{2.5}$ values, for two primary reasons:

- We don't know how Plantower processes the photodiode measurements to calculate the $PM_{2.5}$ mass concentration, but it is not a linear relationship with CH1. Figure S19 shows that the Plantower $PM_{2.5}$ versus $B_{sp}$ shows nonlinear behavior for concentrations below 10–20 µg/m$^3$, that is, comparable to the mean value for Pittsburgh.
- We don't know what mass scattering efficiency of the aerosols was used in other studies.

However, we can estimate the $PM_{2.5}$ mass concentration corresponding to the $B_{sp}$ values in Fig. 13 by assuming a mass scattering efficiency of 2.5 m$^2$/g. The average $PM_{2.5}$ for Pittsburgh of 10 µg/m$^3$ would correspond to a scattering coefficient 25 Mm^-1, which would yield an NRMSE of about 20% in Fig. 13. That is substantially less than the value of 40% that the reviewer mentioned above.

We have created a new Sect. 5.2.7 that uses a portion of Sect. 5.2.5 from the originally submitted manuscript and expands to discuss uncertainties and the relation of the PMS with $PM_{2.5}$. The entirety of this new section is pasted below:

**5.2.7 Relationship between CH1 and PM$_{2.5}$**

The $PM_{2.5}$ mass concentration was not measured by Federal Reference Method (FRM) or Federal Equivalent Method (FEM) instruments at MLO and BOS during this study. Consequently, the PA-PMS $PM_{2.5}$ or CH1 results cannot be compared with $PM_{2.5}$ concentrations, but they can be compared with measured scattering coefficients and discussed in the context of mass scattering efficiency, which ties scattering coefficient to mass concentration.

Figure S19 shows that the PA-PMS $PM_{2.5}$ channel is reasonably well correlated with $b_{sp1}$ for values greater than about 10–20 µg m$^{-3}$, typical of many moderately polluted locations, with a calculated mass scattering efficiency of approximately 2.5 m$^2$ g$^{-1}$.  This value of the mass scattering efficiency is at the low end of the range of values reported by Hand and Malm (2007),

which could reflect the nature of the observed aerosols or an error of the PA-PMS $PM_{2.5}$ mass concentration. This suggests that the effectiveness of the PA-PMS to serve as a $PM^{2.5}$ mass concentration monitor is due both to the sensor behaving like an imperfect integrating nephelometer and to the mass scattering efficiency of ambient $PM_{2.5}$ aerosols being roughly constant with values in the 2–4 $m^2$ $g^{-1}$ range.  However, it is likely that the PA-PMS underestimates $PM_{2.5}$ for very clean areas where $b_{sp1}$ is often less than 10 $Mm^{-1}$. For example, the PA-PMS $PM_{2.5}$ was zero for 1099 of the hours in this study when $b_{sp1}$ was greater than 1 $Mm^{-1}$.

One may obtain a lower bound estimate of the PA-PMS RMSE 1-h average mass concentration from the study results.  Figure 14 shows the PurpleAir scattering coefficient RMSE as a function of the measured scattering coefficient. For example, the PurpleAir NRMSE is 20% for a fine aerosol scattering coefficient of 25 $Mm^{-1}$. For an aerosol having a mass scattering efficiency of 2–3 $m^2$ $g^{-1}$, this is approximately 10 $\mu g$ $m^{-3}$.  Thus, the PurpleAir 1-h average RMSE is roughly 2 $\mu g$ $m^{-3}$. This is somewhat lower than the reported mean absolute error of ~4 $\mu g$ $m^{-3}$ for hourly average $PM_{2.5}$ in Pittsburgh (Malings et al., 2020).  This error assumes that the mass scattering efficiency is fixed and known.  This is generally not the case, and the actual error in the mass concentrations will be larger.

The mean 1-h average fine aerosol scattering coefficient $b_{sp1}$ at MLO during our yearlong study was 1.50 $Mm^{-1}$. From Fig. 14 the PurpleAir had a RMSE of 0.60 $Mm^{-1}$.  For an aerosol having a mass scattering efficiency of 2–3 $m^2$ $g^{-1}$, this corresponds to a 1-h average RMSE of roughly 0.2–0.3 $\mu g$ $m^{-3}$. This is well below the advertised 1-h average MDL of commercial $PM_{2.5}$ monitors. For example, the BAM 1020 specifies a typical hourly detection limit of 3.6 $\mu g$ $m^{-3}$.

We have also added this related paragraph in Sect. 5.2.5:

There are two reasons why the PA-PMS MDL and RMSE values reported in our study are surprisingly low. The TSI 3563 nephelometer has an extremely low detection limit of 0.2 $Mm^{-1}$, which is approximately 1% of Rayleigh scattering. Second, the PA-PMS has very low noise at zero aerosol concentration. If the PA in our study had been collocated with a nephelometer that was not as sensitive as the TSI 3563 in a location having an average fine aerosol coefficient of, say, 30 $Mm^{-1}$, then the PA 1-h average MDL could have been significantly higher than the 0.4 $Mm^{-1}$ we obtained in our study.

9. It would be good to place the results in Sec 5.2.5 PA-PMS size distribution in the context of previous results by Kuula et al., Tryner/Volckens, Zou/May, and He/Dhaniyala.

Kuula et al: https://amt.copernicus.org/articles/13/2413/2020/
Tryner et al: https://doi.org/10.1016/j.jaerosci.2020.105654
Zou et al: https://doi.org/10.1080/02786826.2021.1905148
He et al: https://doi.org/10.1080/02786826.2019.1696015

Our results showed that the PA-PMS size distribution is relatively invariant, despite seeing significant variability in the size distributions measured by the SMPS at Table Mountain. We've added the following statement to Sect. 5.2.6 (formerly Sect. 5.2.5 in the originally submitted manuscript):

In our study, we found that the ambient aerosol size distributions measured with the SMPS varied considerably at Table Mountain, while the Plantower normalized reported size distribution changed very little. Invariant Plantower size distributions were also observed during controlled laboratory studies (Tryner et al., 2020; He et al., 2020; Kuula et al., 2020).

10a. Lines 859-860: It has been shown (e.g. Malings et al.) that the as-reported PM$_{2.5}$ mass concentrations (effectively CH1) are ~double the reference PM$_{2.5}$ values. However, the authors here speculate that the unmodified PurpleAir may underestimate high-RH, low visibility aerosol scattering coefficients. (Is there an unheated nephelometer, given the heat of the halogen lamp?)

In answer to the 'unheated nephelometer' question - Yes - in the IMPROVE network, the National Park Service uses Optec nephelometers, which are open-air nephelometers that measure at ambient conditions, but that's not what was used in this study. There also is now a nephelometer being made by Ecotech (http://ecotech-research.com/aurora-3000) that uses LEDs instead of halogen lamps, which minimizes the heating issue.

10b. Lines 859-860:If I understand correctly, the authors are comparing PA response to scattering coefficient at ambient high RH.

The reviewer is correct that throughout the paper we are comparing the PA measurements to low RH scattering measurements from a nephelometer. However, two points should be considered: (a) at both sites we used the heated PA observations so the PA was at lower RH than ambient, and (b) the ambient RH at both MLO and BOS tends to be quite low. At both sites the ambient RH is below 40% approximately 52% of the time and below 80% approximately 90% of the time. As a result, we seldom were comparing PA response at high humidity to low RH scattering coefficient from the nephelometer.

10c. Lines 859-860: It might help to make this more explicit, e.g. state that "previous studies have reported overestimated PM$_{2.5}$ mass concentrations by the PurpleAir at high RH, but that is when the PA is compared to a reference monitor reporting dry aerosol concentrations - which is quite different than our application of the PA as an aerosol light scattering instrument".

It makes sense that previous studies found PA PM$_{2.5}$ at ambient conditions would be higher than reference method PM$_{2.5}$ at dry conditions, but that's not the point we were trying to make with this statement in the originally submitted manuscript, "Our model predicts that the PMS may not be as responsive to hygroscopic growth as an unheated nephelometer." Rather, the point was that our physical-optical model predicts that the larger aerosol diameter (and different refractive index) response by the Plantower might be reduced relative to a perfect unheated nephelometer, depending in part on aerosol composition. It would be valuable to do controlled experiments with an unheated nephelometer such as the Optec NG2 or Ecotech Aurora 3000 to find if the Plantower does in fact underestimate high RH scattering coefficient, which is why we stated earlier in the paragraph that "future work is planned to compare unheated PA monitors with an unheated nephelometer that does not reduce RH before sampling."

We have not updated the manuscript in response to this comment.

---

## Author Comment (AC2)

We would like to thank the reviewers for their clear and positive reviews.  We have addressed comments below. Our responses are in blue. Changes to the manuscript text where needed are noted in red.

Some things to alert reviewer to:

Abstract is slightly rearranged and has some new text reflecting changes in the manuscript.

We have made some very minor updates to the text to add a clarifying word or sentence in places. These changes are noted in red in the text. We have also made some minor adjustments to some of the figures (e.g., changed um to µm on axes labels).

Added 2 new sections:

- Section 5.2.2 "Predicted aerosol size truncation versus published laboratory data" applies the approach shown in original Sect. 5.2.2 (which is now Sect. 5.2.3) for BOS field observations to laboratory results reported in the literature. This section uses much of the text of the original Sect. 5.2.2 but adds discussion related to comparison with published laboratory results. This section contains a new figure - Fig. 10 in revised manuscript - comparing laboratory observations from literature to our model. Also addresses comment #4.
- Section 5.2.7 "Relationship between CH1 and $PM_{2.5}$" breaks out the brief discussion of this relationship in the original Sect. 5.2.5 and slightly expands it to tie it to mass scattering efficiency information. Also addresses comment #8.

Added cited references to end of supplement.

**Reviewer #2**

The paper "Evaluating the PurpleAir monitor as an aerosol light scattering instrument" by Ouimette et al, examines the possibility of using Purple Air PMS sensor data to determine integrated aerosol light scattering coefficient.  A model considering Mie theory and the sensor geometry is used to predict light scattering signals expected from the sensor and the forward and backward scattering truncation.  The model is used predict sensor performance as a function of particle size and the results confirm that the sensor does not measure size distributions.  And that the signal is proportional to scattering coefficient.

The paper presents a comprehensive picture of the working of PMS5003.  The sensor details, model results and experimental validation adds to the existing knowledge on PMS 5003 and critically confirms findings of other studies that have concluded that the sensor behaves more like a nephelometer rather than a scattering spectrometer.  The paper is well written and its findings are likely to be very useful to the growing community of scientists using these sensors for air quality measurements.

I have minor points for the authors to consider.

(1) Lines 106-108 – "is of light scattered by particles (Kelly et al., 2017) which traditionally has been … using integrating nephelometers".  This sentence should be reworded.  As it reads currently, it seems like light scattering measurements are only made by nephelometers.

Yes, you are right. There are many instruments that use light scattering, such as the Teledyne T640x for FEM PM2.5 and optical particle counters, which use light scattering at discrete angles to derive other particle properties such as mass concentration and size distribution.

We have added the text in red to clarify:

The actual measurement in the PA monitor with its two PMS5003 sensors (PA-PMS), and in many other low-cost aerosol monitors, is of light scattered by particles integrated over a wide range of angles (Kelly et al., 2017), which has traditionally been done in atmospheric research and aerosol monitoring programs using integrating nephelometers.

(2) Lines 137-139.  "Model predictions are then compared with yearlong field data at NOAA's Mauna Loa Observatory …".  Please clarify exactly what predictions are compared with what data.

We have clarified the sentence to say that both model predictions and PMS observations are compared with measurements of aerosol light scattering (at MLO and BOS) and also with measurements of aerosol size distribution (at BOS).  The sentence now reads as

PA-PMS measurements are compared to yearlong measured aerosol light scattering coefficients at NOAA's Mauna Loa Observatory (MLO) in Hawaii and to measured and modeled aerosol light scattering coefficients and aerosol size distribution at the Boulder Table Mountain (BOS) site in Colorado.

(3) Lines 140-141: "… an empirical relationship is developed to estimate the light scattering and uncertainty from the PA-PMS data."  Light scattering intensity?  And uncertainty of what?

We have clarified the sentence as follows:

Finally, an empirical relationship is developed to estimate the submicron light scattering coefficient and its uncertainty from the PA-PMS data.

(4) How is the uncertainty in the physical geometry and optical geometry accounted for in the model?

The originally submitted manuscript did not include how the uncertainty in the physical geometry and optical geometry is accounted for in the model. The revised paper now includes this uncertainty in geometry on model predictions in a new table - Table S3 (pasted

below). We have added the following text in Sect. 3.3 and Sect. 5.2.1 to further address this:

The variance in the PMS physical and optical geometry and errors in the measurements are not known but likely small. To evaluate the sensitivity of the modeled PA scattering to errors in these measurements, the model was exercised with large deviations of ±25% and ±50% in these inputs. As shown in Table S3, the errors tend to increase with particle size. The modeled PA scattering to a perfect nephelometer is most sensitive to errors in the distance from the laser to the photodiode. For particle diameters of 0.5 μm, +25% and +50% changes in this distance resulted in maximum differences of 10% and 20%, respectively. Based on these results and the fact that the errors in the physical dimensions are less than 25%, these errors are thought to have a small contribution to the overall modeled PA scattering error and were not directly accounted for in the analysis. This analysis does not attempt to account for the possibility that the laser beam profile is not a simple plane wave or that the laser beam profile may evolve significantly as it is focused over the photodiode, and the standard plane wave Mie calculations would no longer apply.

Table S3 added to Sect. S4:

**Table S3**. Effect of uncertainty in measurement of the PMS geometry on model predictions of scattering ratio compared to a perfect nephelometer as a function of particle diameter. The % changes in various dimensions (left most column) are compared to the base case predictions. The base case dimensions are in Sect. 2.2.4, and the base case predictions are on Fig. 7.

| Particle diameter (μm) | 0.30 | 0.50 | 0.70 | 1.0 | 2.0 | 4.0 |
|---|---|---|---|---|---|---|
| Scattering ratio for the base case geometry | 0.88 | 0.55 | 0.38 | 0.31 | 0.22 | 0.23 |
| Change in distance from laser to photodiode (%) | Percent change in scattering ratio compared to base case | | | | | |
| -50 | -3 | 4 | 26 | 42 | 39 | 26 |
| -25 | 0 | 7 | 20 | 24 | 13 | 15 |
| 25 | -2 | -10 | -21 | -17 | -4 | -10 |
| 50 | -3 | -20 | -36 | -26 | -10 | -21 |
| Change in diameter of exposed photodiode (%) | Percent change in scattering ratio compared to base case | | | | | |
| -50 | -4 | -10 | -26 | -14 | -7 | -12 |
| -25 | -2 | -5 | -10 | -9 | -4 | -4 |
| 25 | 1 | 4 | 8 | 9 | 4 | 5 |
| 50 | 2 | 6 | 13 | 16 | 8 | 9 |
| Change in distance of laser exit hole to photodiode (%) | Percent change in scattering ratio compared to base case | | | | | |
| -50 | -2 | -8 | -25 | -19 | -2 | -8 |

| | | | | | | |
|---|---|---|---|---|---|---|
| -25 | -2 | -2 | -7 | -11 | 0 | -3 |
| 25 | 0 | -7 | -2 | 2 | -3 | -4 |
| 50 | 0 | -12 | -5 | 3 | -4 | -6 |
| Change in thickness of base mask over the photodiode (%) | Percent change in scattering ratio compared to base case | | | | | |
| -50 | 1 | 2 | 1 | 0 | 3 | 1 |
| -25 | 0 | 1 | 1 | 0 | 1 | 1 |
| 25 | -1 | -2 | -1 | -1 | -2 | -2 |
| 50 | -1 | -5 | -4 | -3 | -5 | -5 |
| Change in distance from photodiode to light trap (%) | Percent change in scattering ratio compared to base case | | | | | |
| -25 | 3 | 3 | 3 | 3 | -2 | 1 |
| -50 | 1 | 1 | 1 | 1 | -2 | 0 |
| -75 | 3 | 3 | 3 | 3 | -2 | 1 |

Additionally we have added the following text to Sect. 5.2.1:

The predicted photodiode output is linearly correlated with the ordinary least squares (OLS) regression ($R^2 = 0.90$, normalized root mean square error (NRMSE) ~25%) with CH1 over 4 orders of magnitude. The RMSE contains contributions of errors from the model-predicted radiant power, the measured SMPS data the model is based on, as well as in the CH1 measurements. This strong correlation and low RMSE is convincing evidence that the model and SMPS data describe the PMS response quite well.

(5) In Figure 2, the precision is shown as a function of concentration. How much of the decrease in precision with decreasing concentrations can be explained by Poisson statistics of number of particles expected in the viewing volume of the units?

We have added the following text in Sect. 2.2.9 to address this comment:

There are two mechanisms that may contribute to the rapid uncertainty increase for CH1 < 100. First, it is likely that some of the increased uncertainty in CH1 below values of 100 is inherent to sampling low concentrations, as is the case for any instrument. Second, the geometry of the laser sensing volume in the PMS can contribute to uncertainty in the CH1 at low concentrations, specifically if particles are not distributed uniformly within the laser beam.

 (6a) Figure 10: the x-axis scale is unusual – please use linear or log-scale.

Figure 10 is now Figure 11 in the revised manuscript. The bin spacings on the original graph were just the SMPS size channels. The x-axis has been replaced by a logarithmic scale of MSD values where the upper and lower bin values are selected as MSD$_i$+MSD$_{i+1}$/2 and

MSD$_i$-MSD$_{i-1}$/2 where i refers to the i$^{th}$ bin.  The midpoints of the bins are 0.2239, 0.2512, 0.2818, 0.3162, 0.3548, 0.3981, 0.4467, 0.5012, 0.5623, 0.6310, 0.7079, and 0.7943. Although the MSD values were selected based on a log scale, they are plotted equally spaced from each other to maintain uniformity in the dimensions of the box and whisker symbols. The solid blue horizontal lines, which correspond to the number of observations in each bin, have been added for clarity.

We have added some more explanatory text, updated the figure to include points in each bin, and modified the caption to explain the unusual scale.  This is the new text, figure, and figure caption:

The results are shown in Fig. 11 as a box and whisker plot of the CH1avg/b$_{sp1}$ values found in each MSD bin.  The center MSD value for each bin is based on a logarithmic scale of MSD values where the upper and lower bin values are selected as MSD$_i$+MSD$_{i+1}$/2 and MSD$_i$-MDS$_{i-1}$/2 where i refers to the i$^{th}$ bin. The thin black horizontal lines correspond to the number of observations in each bin and the scale is shown on the right hand axis. There are less than 20 values in the 0.22 µm, 0.63 µm, 0.71 µm, and 0.79 µm bins.  Approximately 67% of the MSDs observed at BOS were between 0.29 µm and 0.36 µm, and 98% of MSDs were between 0.26 µm and 0.46 µm. The overall average CH1avg/b$_{sp1}$ ratio, based on 6777 observations, is 65 Mm.

[Figure]

**Figure 11.** Observed decrease in CH1avg/b$_{sp1}$ ratio as a function of MSD values.  MSD values were selected based on a log scale but plotted equally spaced from each other to maintain uniformity in the dimensions of the box and whisker symbol.  Red line represents the median value, and the bottom and top of each box are the first and third quartile values.  Extremes shown

on each box are the 2 and 98 percentiles.  Black horizontal lines for each MSD value are the number of observations in the respective MSD bins.

(6b) Also, could these results be compared against model predictions as validation of model performance?

A comparison of modeled to measured PA response to the measured aerosol distribution is shown and discussed in Fig. 9.

(7) Section 4.5:  It would be good to add a sentence or two about how the nephelometer was integrated with the DMPS for aerosol scattering coefficient distribution measurements.

We've expanded the sentence describing the scattering calculation from the DMPS as follows (new text is in red):

The 0.1 μm to 0.8 μm channels of the DMPS were used to calculate hourly-average fine aerosol scattering coefficient distributions and the total fine aerosol scattering coefficient, assuming spherical particles (Mie theory) with a refractive index of 1.53 - 0.017i.